

# A Physics-Constrained Deep-Learning Framework based on Long-Term Remote-Sensing Data for Retrieving Vertical Distribution of PM$_{2.5}$ Chemical Components

Hongyi Li[1], Ting Yang[1*], Yele Sun[1], Zifa Wang[1, 2]

[1] State Key Laboratory of Atmospheric Environment and Extreme Meteorology, Institute of Atmospheric Physics, Chinese Academy of Sciences, Beijing 100029, China
[2] College of Earth and Planetary Sciences, University of Chinese Academy of Sciences, Beijing 100049, China

*Correspondence to*: Ting Yang (tingyang@mail.iap.ac.cn)

**Abstract.** The vertical distribution of PM$_{2.5}$ chemical components is crucial for identifying the causes of atmospheric pollution and its impact on climate change and extreme weather. By integrating long-term lidar measurements, deep-learning algorithms and a physics-constrained optimization method, this paper presents a novel lidar-based retrieval framework to obtain vertical mass concentration profiles of PM$_{2.5}$ chemical components for the first time. Identifiable components include sulfate (SO$_4^{2-}$), nitrate (NO$_3^-$), ammonium (NH$_4^+$), organic matter (OM) and black carbon (BC), which extend beyond the component types that traditional remote-sensing retrievals can identify. A 1-year retrieved surface mass concentrations of these components closely aligned with the observations, with Pearson correlation coefficient values ranging from 0.91 to 0.98. The retrieval framework applied in varying non-training spatiotemporal scenarios also showed robust generalization capabilities. Tower and aircraft-based field campaigns indicate that the retrieved and observed vertical profiles of these components exhibited consistent patterns in mass concentrations and proportions. Subsequently, an explainable method was incorporated into the retrieval framework to quantify the multivariate driving effects on vertical profile retrieval. Results showed that the extinction coefficient and representative indicators within physiochemical processes contributed significantly to mass concentrations of these components. Finally, a dataset of vertical mass concentration profiles of these components over six years in a Chinese megacity was generated by the retrieval framework, revealing the dominant roles of OM and NO$_3^-$ in PM$_{2.5}$ throughout the entire boundary layer across all seasons. Through implementing clean air policies, the reduction rates of these components in the megacity exhibited the highest reduction rate of 0.17-0.82 µg m$^{-3}$ a$^{-1}$ occurring at an altitude of ~300 m. Our retrieval framework offers a novel approach for acquiring vertical profiles of PM$_{2.5}$ chemical components, thereby providing a new perspective on elucidating the vertical evolution of atmospheric pollutants.

## 1 Introduction

PM$_{2.5}$ is a complex mixture composed of varying chemical components (Tao et al., 2017), mainly including sulfate (SO$_4^{2-}$), nitrate (NO$_3^-$), ammonium (NH$_4^+$), organic matter (OM) and black carbon (BC). The diverse physiochemical properties



arising from various chemical components yield distinct effects on the environment (Tan et al., 2018), climate change (Menon et al., 2002; Zhu et al., 2024) and human health (Kim et al., 2022). Vertical detection technologies have revealed that chemical components are primarily distributed at varying heights within the atmospheric boundary layer and contribute to environmental pollution through internal physiochemical processes (Morgan et al., 2009; Yang et al., 2024; Sun et al., 2015). Additionally, the proportion and vertical distribution of chemical components can regulate radiation flux at both the

top of the atmosphere and at the surface by directly affecting light absorption and scattering, as well as the microphysical properties of clouds, thereby influencing climate change and extreme weather (Zhao et al., 2024). Consequently, characterizing the vertical structures of chemical components is essential for identifying the causes of $PM_{2.5}$ pollution and the response mechanisms related to climate change and extreme weather.

Field campaigns are widely conducted to obtain vertical profiles of $PM_{2.5}$ chemical components by mounting observation instruments on meteorological towers (Lei et al., 2021), aircraft (Liu et al., 2019), tethered balloons (Babu et al., 2011) and unmanned aerial vehicles (Jiang et al., 2022). However, these platforms are constrained by sparse detection sites and heights, limited flight schedules, and high observation costs (Dubey et al., 2022), hindering the time-continuous acquisition of vertical profiles of $PM_{2.5}$ chemical components within the whole boundary layer over a long-term period. Continuous

remote-sensing lidar detection technologies with high temporal and vertical resolution serve as robust pathways for the constant identification of $PM_{2.5}$ components across all altitudes (Wang et al., 2022). Additionally, both satellite-based lidar and ground-based lidar networks, such as China Lidar Joint Observation Network (LiDARNET, https://lidar.pku.edu.cn/, last access: 25 July 2025), Asian Dust Network (AD-NET) (Sugimoto et al., 2005), Micro Pulse Lidar Network (MPLNET) (Welton et al., 2001), and European Aerosol Research Lidar Network (EARLINET) (Ansmann et al., 2003), provide remote

sensing capabilities with extensive spatial coverage.

Retrieval algorithms for the lidar have been progressively developed over the past 20 years. Earlier studies utilized lidar depolarization ratios to identify dust and non-dust aerosols (Sugimoto et al., 2003; Tesche et al., 2009). Subsequently, additional lidar parameter constraints, such as multi-wavelength backscatter coefficient and lidar ratio, were incorporated to

identify dust aerosol, water-soluble aerosols, black carbon, and sea salt based on the assumption of external mixing (Nishizawa et al., 2011; Nishizawa et al., 2017). Hara et al. (2018) considered the hygroscopic growth of water-soluble aerosols and their internal mixing with BC to mitigate the overestimation of BC retrieval (Hara et al., 2018). By integrating the ground-based lidar and sun-photometer, Wang et al. (2022) significantly increased the identifiable aerosol component types, including ammonium nitrate-like, water-insoluble organic matter, water-soluble organic matter, black carbon and fine-

mode aerosol water content (Wang et al., 2022). However, the aerosol component type retrieved from existing lidar retrieval algorithms that utilize aerosol optical properties is not equivalent to the conventional chemical component type. Due to similar optical properties exhibited by $PM_{2.5}$ chemical components, the identification of chemical component types seems to





be beyond the scope of remote-sensing retrieval (Wang et al., 2022). Moreover, the multiple parameterization assumptions introduced by existing lidar retrieval algorithms increase the uncertainties in component retrieval.

Data-driven machine learning can interpret the nonlinear relationships between PM$_{2.5}$ chemical components and various driving factors without the constraints imposed by the inherent properties of these components (Li et al., 2025a). Meng et al. (2018) utilized a random forest algorithm to predict national mass concentrations of SO$_4^{2-}$, NO$_3^{-}$, organic carbon (OC) and elemental carbon (EC), achieving R$^2$ values ranging from 0.71 to 0.86 on a daily scale (Meng et al., 2018). Based on this

algorithm, Lv et al. (2021) further achieved the hourly predictions of the aforementioned chemical components and NH$_4^{+}$ with R values of 0.71-0.81 (Lv et al., 2021). Subsequently, deep learning algorithms are employed to accurately characterize complex nonlinear relationships and effectively extract data features, thereby enhancing the predictive ability of hourly mass concentrations of PM$_{2.5}$ chemical components (Lee et al., 2023; Liu et al., 2023; Li et al., 2025a). However, current studies primarily focus on predicting the ground-level mass concentrations of PM$_{2.5}$ chemical components but cannot interpret the

vertical distribution of these components. Furthermore, existing prediction models are susceptible to the quantity and quality of available training data due to the absence of physical constraints, limiting their spatiotemporal generalization capabilities.

In this study, we proposed a novel physics-constrained deep-learning framework that utilized lidar data to retrieve vertical profiles of five PM$_{2.5}$ chemical components (SO$_4^{2-}$, NO$_3^{-}$, NH$_4^{+}$, OM and BC) for the first time. Our retrieval framework

effectively mitigates the limitations of remote-sensing retrieval algorithms in identifying chemical components, as well as the deficiencies and limited generalization capabilities of purely data-driven machine learning techniques in characterizing vertical profiles of these components. Detailed descriptions of the retrieval framework and the data utilized are provided in Sect. 2., while Sect. 3 discusses the validation of the retrieval framework, the assessment of feature importance, and applications of this framework. Section 4 presents the conclusion.

**2 Methodology and data**

**2.1 Methodology**

**2.1.1 Retrieval framework**

This paper proposed a novel retrieval framework for retrieving vertical distribution of five PM$_{2.5}$ chemical components (NH$_4^{+}$, SO$_4^{2-}$, NO$_3^{-}$, OM and BC) by integrating long-term lidar measurements, deep learning algorithms, hyperparameter

tuning, and physics-constrained optimization for the first time (Fig. 1). The aerosol extinction coefficient at 532 nm ($\sigma_{bsc,532}$) and multiple meteorological parameters (u-component wind, v-component wind, temperature, relative humidity, specific humidity, vertical velocity and geopotential) serve as input features. The deep learning module (Fig. 1, red part), mainly consisting of Convolutional Neural Network (CNN), Bidirectional Long Short-Term Memory (BiLSTM), attention



mechanism and Bayesian hyperparameter optimization, is utilized to establish the nonlinear relationship between input
features and five $PM_{2.5}$ chemical components. Notably, the input and output data are normalized by Z-score normalization to
stabilize the training process, speed up training convergence, and enhance model robustness (Al-Faiz et al., 2018; Cabello-
Solorzano et al., 2023). Therefore, the vertical profiles of five $PM_{2.5}$ chemical components retrieved by the deep learning
module are denormalized by available aircraft-based vertical observations (Fig. 1, yellow part). A physics-constrained
optimization module in this framework is designed to reduce spatiotemporal representativeness errors in limited aircraft-
100 based vertical observations during the denormalized process. The physics-constrained optimization module incorporates a
multi-object loss function based on the Interagency Monitoring of Projected Visual Environment (IMPROVE) Equation with
Non-dominated Sorting Genetic Algorithm II (NSGA-II). The detailed description of deep learning algorithms,
hyperparameter tuning, and physics-constrained optimization will be presented below.

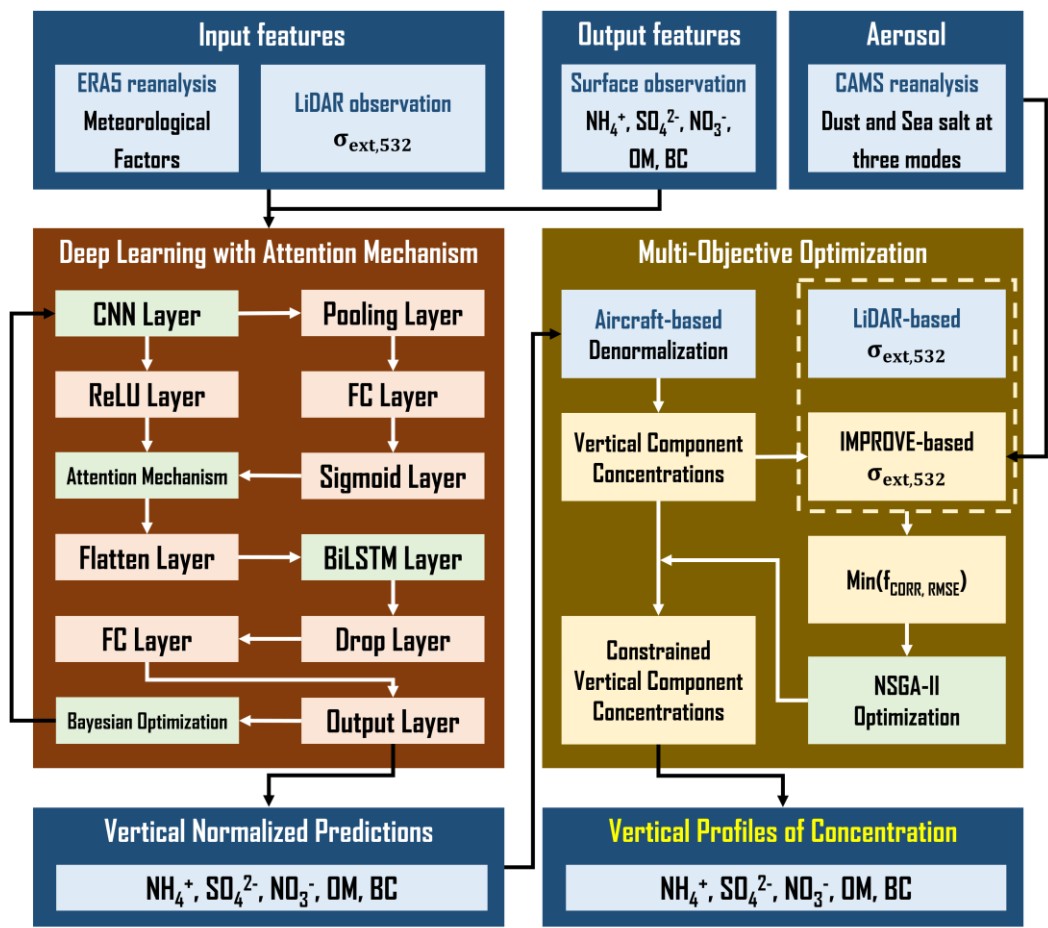

**Figure 1: Remote-sensing retrieval framework for vertical distribution of five $PM_{2.5}$ chemical components ($NH_4^+$, $SO_4^{2-}$, $NO_3^-$, OM and BC).**



### 2.1.2 Deep learning and hyperparameter tuning

The deep learning module is the core of the retrieval framework that generates the normalized vertical profiles of five PM$_{2.5}$

chemical components by feeding the lidar-based aerosol extinction coefficient at 532 nm ($\sigma_{ext,532}$) and multiple meteorological parameters. The deep learning module is designed by numerous neural network layers (Fig. 1, red part), including the CNN layer, Average Pooling layer, Rectified Linear Unit (ReLU) layer, Fully Connected (FC) layer, Attention Mechanism layer, Sigmoid layer, Flatten layer, BiLSTM layer, Dropout layer and Regression Output layer. The CNN, Attention Mechanism and BiLSTM layers are the key layers to capture the multivariate and temporal characteristics,

contributing to the nonlinear relationship between the input and output features. Other layers are responsible for data input, structural transformation, normalization, nonlinear process, pooling process, neuron removal and data output, enhancing the training performance and preventing overfitting. Here, we review the description of three key layers, and the description of other layers can be found in our previous work (Li et al., 2025a).

CNN is a variant of the multilayer perceptron that efficiently identifies the relevant features through local perception, sparse connections and sharing of weight and bias (Alzubaidi et al., 2021). The convolutional layer in CNN performs convolutional computation on input across spatial dimensionality using learnable kernels to extract local features and enhance training efficiency (O'shea and Nash, 2015). Then the convolutional output is typically enhanced nonlinearly by the ReLU layer (Eq. (1)) or down-sampled nonlinearly by the pooling layer in a CNN architecture.

$$y_t = max\big(0, f(\mathbf{w} \times x_t + b_t)\big), \tag{1}$$

Where $y_t$ is the nonlinearly enhanced convolutional output at timestep $t$, $f(w \times x_t + b_t)$ is the original convolutional output at timestep $t$, $x_t$ is the input data at timestep $t$, $w$ is the weight matrix and $b_t$ is the bias term.

The attention mechanism layer is incorporated with CNN to amplify the weight of key information and mitigate the

interference of redundant information, leading to an enhancement in the quality of the CNN output (Wang and Zhang, 2025). The attention mechanism is inspired by the ability of human vision to selectively focus on key information (Guo et al., 2022). Our retrieval framework integrates this layer by dynamic weight allocation to enhance the importance of key features and reduce the interference of irrelevant features. The weight is generated by the FC layer (Eq. (2)) and performs Schur product operation with CNN multivariate output (Eq. (3)).

$$\mathbf{W} = \text{sigmoid}\Big(\text{FC}\big(\text{Pooling}(y_{1,i})\big)\Big), \tag{2}$$

$$y_{2,i} = \mathbf{W} \cdot y_{1,i}, \tag{3}$$



Where $y_{1,i}$ is the CNN multivariate output. Pooling and FC layers are responsible for down-sampling and feature learning, respectively, thus predicting the importance of $i^{th}$ feature. The sigmoid activation function is utilized to calculate the attention weight (**W**). $y_{2,i}$ is the reweighted multivariate output.

BiLSTM is a variant of Recurrent Neural Networks (RNNs) that learns long-timestep information bidirectionally and avoids the gradient vanishing or explosion of traditional RNNs (Kavianpour et al., 2023). Previous studies have indicated that BiLSTM outperforms LSTM in regression tasks due to the insufficient utilization of future information in LSTM (Siami-Namini et al., 2019; Yang and Wang, 2022). Therefore, the BiLSTM layer is integrated into the deep-learning module to

fully capture the temporal characteristics of the CNN attention-weighted multivariate output. The BiLSTM layer is realized by the forward LSTM and backward LSTM (Eq. (4)). Both the forward and backward LSTM consist of cell states, forget gates, input gates, output gates, and activation functions, which are responsible for transmission, screening and processing of temporal information. The final LSTM output is obtained by output gates and cell states (Eq. (5)). A detailed description of BiLSTM components can be found in our previous work (Li et al., 2025a).

$$H_t = [\vec{h}_t; \overleftarrow{h}_{end-t+1}],  \tag{4}$$

$$h_t = o_t \times \tanh(C_t),  \tag{5}$$

Where $H_t$ is the final output of BiLSTM at timestep $t$, which is obtained by concatenating the forward output $\vec{h}_t$ and backward output value $\overleftarrow{h}_{end-t+1}$. $h_t$ is the final output of LSTM at timestep $t$, $o_t$ is the output of output gate at timestep $t$, tanh is an activation function that regulates the values transmitted in neural networks by compressing the values to a range

of from -1 to 1. $C_t$ is the output of the cell state at timestep $t$.

Hyperparameter tuning is crucial for improving the performance of deep neural networks. Bayesian optimization can determine global optima with higher efficiency (Shahriari et al., 2016) and has been widely employed in hyperparameter optimization of varying machine learning models (Wu et al., 2019a). The primary process of Bayesian optimization involves

establishing search spaces for hyperparameters and the corresponding objective function, followed by the determination of the optimal solution by minimizing the objective function (Eq. (6)). Bayesian optimization utilizes a probabilistic surrogate model to iteratively estimate the complex unknown objective function based on the current query point and then identifies the next most promising query point by an acquisition function (Shahriari et al., 2016). The probabilistic surrogate model and the acquisition function in this study are the Gaussian process regression model (Rasmussen, 2004) and the Expected-

Improvement-Per-Second-Plus function (Gelbart et al., 2014), respectively.

$$x^* = argmin\, f(x), x \in X \subseteq R^d,  \tag{6}$$

Where $x^*$ is the optimal scheme of multiple hyperparameters, $x$ is the decision vector composed of d-dimensional hyperparameters, $X$ is the search space that consists of all possible decision vectors, $f(x)$ is the unknown objective function.



### 2.1.3 Physics-constrained optimization scheme

The normalized vertical profiles of PM$_{2.5}$ chemical components generated by the deep learning module are denormalized by the statistical characteristics of winter aircraft measurements with an altitude ranging from 100 m to 2900 m in Beijing (Liu et al., 2020). To reduce the spatiotemporal representativeness error induced by the statistical results of aircraft measurements, we initially scaled the aircraft-based observations of PM$_{2.5}$ chemical components based on the ratio of in-situ PM$_{2.5}$ observations at the specified location and period to aircraft-based PM$_{2.5}$ observations. Subsequently, a physics-constrained

optimization scheme is incorporated into the retrieval framework based on a revised Interagency Monitoring of Projected Visual Environment (IMPROVE) Equation (Pitchford et al., 2007) and Non-dominated Sorting Genetic Algorithm II (NSGA-II) (Verma et al., 2021).

The revised IMPROVE Equation interprets the particle extinction coefficient ($\sigma$) through the concentrations and the optical

and microphysical characteristics of PM$_{2.5}$ chemical components (Eq. (7)).

$$\sigma = \theta_s^{SNA} f(RH)[SO_4^{2-} + NO_3^- + NH_4^+] + \theta_s^{OC}[OC] + \theta_s^{FS}[Fine\ Soil] + \theta_s^{CM}[Coarse\ Mass] +$$
$$\theta_s^{FSS} f_{FSS}(RH)[Fine\ Sea\ Salt] + \theta_a^{BC}[BC] + Rayleigh\ Scattering, \tag{7}$$

Where $\sigma$ is the estimated particle extinction coefficient (km$^{-1}$), $\theta_s$ is the scattering efficiency (m$^2$ mg$^{-1}$), $\theta_a$ is the mass absorption efficiency (m$^2$ mg$^{-1}$), respectively. $f(RH)$ and $f_{FSS}(RH)$ account for the increase in light scattering induced by

hygroscopic growth of sulfate, nitrate and ammonium (SNA), as well as fine sea salt (FSS). $\theta_s^{FS}$, $\theta_s^{CM}$, $\theta_s^{FSS} f_{FSS}$ and $\theta_a^{BC}$ are set to 0.001 m$^2$ mg$^{-1}$, 0.0006 m$^2$ mg$^{-1}$, 0.0017 m$^2$ mg$^{-1}$ and 0.01 m$^2$ mg$^{-1}$, respectively. $Rayleigh\ Scattering$ is set to 0.01 km$^{-1}$. $\theta_s^{SNA}$ and $\theta_s^{OC}$ are determined by Eq. (8)-(9).

$$\theta_s^{SNA} = 0.003 \times (0.7 + 0.002 \times [SO_4^{2-} + NO_3^- + NH_4^+ + OC]), \tag{8}$$
$$\theta_s^{OC} = 0.00363 \times (0.7 + 0.002 \times [SO_4^{2-} + NO_3^- + NH_4^+ + OC]), \tag{9}$$

Then we employ the Pearson correlation coefficient (CORR) and root mean square error (RMSE) between estimated and observed extinction coefficients to establish a multi-objective function that regulates the denormalized vertical profiles of PM$_{2.5}$ chemical components by a scale factor (Eq. (10)). The NSGA-II algorithm is utilized to determine the optimal scale factor by solving a multi-objective function that simultaneously enhances the correlation and reduces the discrepancy

between the estimated and observed extinction coefficients (Eq. (11)).

$$M_{regulated}^{i,h} = \gamma_{i,h} \times M_{original}^{i,h}, i = SO_4^{2-}, NO_3^-, NH_4^+, OM, and\ BC, \tag{10}$$
$$\gamma_{i,h} = \min(f_{RMSE}(\gamma), f_{CORR}(\gamma)), \tag{11}$$

Where $M_{regulated}^{i,h}$ (µg m$^{-3}$) is the regulated mass concentration of the i$^{th}$ chemical component at an altitude of $h$ (m), $\gamma_{i,h}$ is the scale factor for the i$^{th}$ chemical component at an altitude of $h$ (m), and $M_{original}^{i,h}$ (µg m$^{-3}$) is the original mass concentration of the i$^{th}$ chemical component at an altitude of $h$ (m).

concentration of the i$^{th}$ chemical component at an altitude of $h$ (m).



NSGA is capable of simultaneously optimizing the multi-objective function by generating a Pareto front that consists of an ensemble of non-dominated solutions (Srinivas and Deb, 1994). The non-dominated solutions in a Pareto front meet the criterion that one objective cannot be further improved without compromising other objectives. However, the initial version

of NSGA has several limitations. First, NSGA has a high computational complexity of $O(MN^3)$, where M is the number of objective functions, and N is the size of the population. Second, NSGA utilizes a sharing parameter to preserve the diversity of the population that dominates the choice of Pareto non-dominated solutions, resulting in the introduction of parameter uncertainty into the algorithm. Third, NSGA lacks an elitism mechanism, leading to the incorrect removal of advantageous solutions. NSGA-II is an improved NSGA with a lower computational complexity of $O(MN^2)$ and an elitism mechanism

that retains the dominant members of the parent and offspring generations during iterative evolution (Deb et al., 2002). Moreover, NSGA-II replaces the sharing parameters in NSGA with the crowding distance operator, mitigating the uncertainty of sharing parameters and the high computational complexity of sharing functions.

NSGA-II implements multi-objective optimization by two primary procedures, namely non-dominated sorting and crowding

distance calculation. The non-dominated sorting progressively identifies the Pareto front at each rank from a population of size N. The Pareto front at the second rank is derived from a population that excludes the Pareto front at the first rank. The crowding distance is utilized to quantify the priority of all optimal solutions within a Pareto front, defined as the normalized distance of two nearest optimal solutions on either side (Eq. (12)).

$$d_i = \sum_{i=1}^{K} \frac{f_m^{i+1} - f_m^{i-1}}{f_m^{max} - f_m^{min}}, \tag{12}$$

Where $d_i$ is the crowding distance of the $i^{th}$ intermediate Pareto optimal solution, $K$ is the number of Pareto optimal solutions in a Pareto front, $f_m^{i+1}$ is the $m^{th}$ objective value induced by the $(i+1)^{th}$ Pareto optimal solution, $f_m^{i-1}$ is the $m^{th}$ objective value induced by the $(i-1)^{th}$ Pareto optimal solution, $f_m^{max}$ and $f_m^{min}$ are the $m^{th}$ maximum and minimum objective values, respectively.

The workflow of NSGA-II is summarized as follows (Fig. 2).

a) Randomly generating an initial population ($A_1$) of size N. Performing selection, crossover and mutation operations on $A_1$ to generate an offspring population ($B_1$) of size N. The parent population ($A_1$) and the offspring population ($B_1$) are combined to form a new population ($C_1$) of size 2N.

b) Performing a rapid non-dominated sorting on $C_1$ to generate the Pareto fronts ($P_i$, $i = 1, 2, ..., n$) at different ranks.

c) Filling the next population ($A_2$) of size N with $P_i$ based on the rank order.

d) When $A_2$ is filled to the point of insufficient capacity to contain the entire $P_i$, the optimal solutions in $P_i$ are inserted into $A_2$ in a priority order identified by the non-dominated sorting and crowding distance until the size of $A_2$ reaches N.



e) Performing selection, crossover and mutation operations on $A_2$ to generate an offspring population ($B_2$) of size N. The parent population ($A_2$) and offspring population ($B_2$) combine to form a new population ($C_2$) of size 2N.

f) Iterating steps (b) to (e) until the convergence criteria are satisfied.

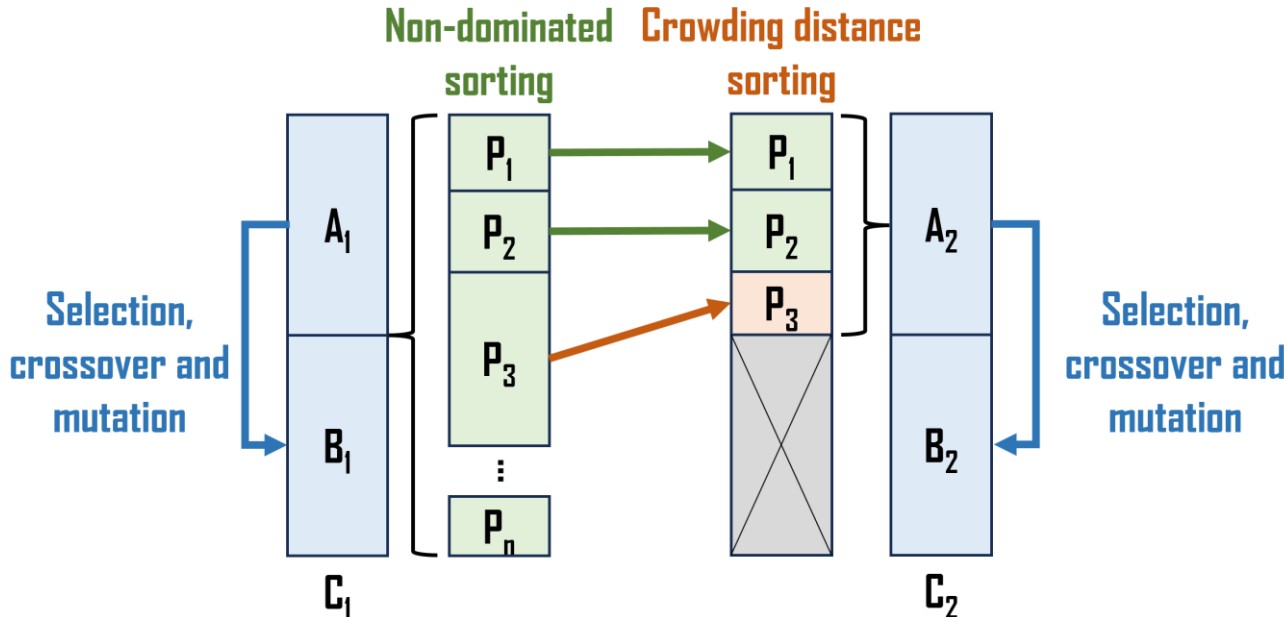

**Figure 2: Brief workflow of NSGA-II.**

**2.1.4 Framework training and evaluation**

A long-term (2021-2022) hourly multivariate dataset that contains $\sigma_{bsc,532}$, u-component wind, v-component wind, temperature, relative humidity, specific humidity, vertical velocity, geopotential, $NH_4^+$, $SO_4^{2-}$, $NO_3^-$, OM and BC, was employed to train the deep learning module, with 80% of the dataset randomly allocated for training and 20% reserved for testing. The iteration number of Bayesian optimization is set to 20.

To fully evaluate the performance of the retrieval framework in predicting vertical profiles of $NH_4^+$, $SO_4^{2-}$, $NO_3^-$, OM and BC, we conduct three retrieval experiments: (1) We compare the retrieved mass concentrations with the observed values at the surface level during a training year (2021) and three non-training years (2017, 2018 and 2024). (2) We assess the spatial generalization ability by applying the retrieval framework to 39 non-training lidar sites in the Beijing-Tianjin-Hebei (BTH) region on February 8-15th, 2021 and comparing the retrieved mass concentrations with observations at the surface level. (3)

We validate the retrieved vertical profiles by aircraft-based and tower-based vertical observations during several non-training episodes. Subsequently, SHapley Additive exPlanations (SHAP), a local explainable technology (Lundberg et al., 2020), has



been widely employed in prediction interpretation for varying machine learning models (Li et al., 2025b; Hou et al., 2022), is integrated into the deep learning module to quantify the impact of multivariate input features on the retrieval of $PM_{2.5}$ chemical components. Finally, we applied this retrieval framework to generate a long-term vertical profile dataset for five

$PM_{2.5}$ chemical components in a megacity over six years of 2017-2018 and 2021-2024.

## 2.2 Data

### 2.2.1 Lidar measurement

The $\sigma_{bsc,532}$ data for deep learning module training and $PM_{2.5}$ chemical component retrieving is obtained from a ground-based dual-wavelength polarization Mie lidar at the Institute of Atmospheric Physics (IAP), Chinese Academy of Sciences

(CAS), Beijing (39.98°N,116.38°E). This Mie lidar has consistently detected optical signals since 2017, offering a temporal resolution of 15 minutes and a vertical resolution of 6 m. The $\sigma_{bsc,532}$ data on February 8-15th, 2021 at 39 BTH lidar sites for spatial generalization ability assessment are provided by the China National Environmental Monitoring Center (CNEMC). The multi-site data offers a temporal resolution of 5-20 minutes and a vertical resolution of 7.5 m. These minute-level resolution data are averaged to achieve an hourly resolution.

### 2.2.2 Auxiliary data for Retrieval

The data of multiple meteorological parameters for deep learning module training and $PM_{2.5}$ chemical component retrieving can be obtained from the 5th Generation European Centre for Medium-Range Weather Forecasts (ECMWF) ReAnalysis (ERA5, https://cds.climate.copernicus.eu/datasets, last access: 25 July 2025), which provides the hourly data on pressure levels (1000-1 hPa) from 1940 to present with a spatial resolution of 0.25° × 0.25°. The data of fine soil, coarse mass and

fine sea salt for physics-constrained optimization can be obtained from 4th Generation ECMWF Atmospheric Composition Reanalysis (EAC4, https://ads.atmosphere.copernicus.eu/datasets, last access: 25 July 2025), which provides the 3-hourly data on pressure levels (1000-1 hPa) from 2003 to 2024 with a spatial resolution of 0.75° × 0.75°. The mass concentration (μg m$^{-3}$) of fine soil is approximately estimated by the mixing ratio (kg kg$^{-1}$) of dust aerosol with a diameter of 0.03-0.9 μm. The mass concentration (μg m$^{-3}$) of coarse mass is approximately estimated by the mixing ratio (kg kg$^{-1}$) of dust aerosol with

a diameter of 0.9-20 μm. The mass concentration (μg m$^{-3}$) of fine sea salt is approximately estimated by the mixing ratio (kg kg$^{-1}$) of sea salt aerosol with a diameter of 0.03-5 μm. The pressure levels (hPa) of ERA5 and EAC4 are converted to geometric heights (m), and the 3-hour EAC4 data is converted to hourly data through a linear interpolation method. The grid cells of EAC4 and ERA5 that contain the lidar sites were extracted using the k-nearest neighbor search method based on longitude and latitude data (Friedman et al., 1977).





### 280 2.2.3 Surface observations

Ground-level mass concentrations of $NH_4^+$, $SO_4^{2-}$, $NO_3^-$, OM and BC at the Beijing lidar site (39.98°N,116.38°E) were collected for training the deep learning module and validating retrievals by a high-resolution time-of-flight aerosol mass spectrometer, with a temporal resolution of 1 hour, covering the periods from January 1, 2021, to March 31, 2022, and June 1 to August 31, 2022. Ground-level mass concentrations of the five $PM_{2.5}$ chemical components at 39 non-training BTH sites 285 were provided by CNEMC. Besides, ground-level $PM_{2.5}$ mass concentrations, approximately equal to the sum of the mass concentrations of the five chemical components, are available on CNEMC data release website (https://www.cnemc.cn/, last access: 25 July 2025).

### 2.2.4 Aircraft-based and tower-based measurements

The aircraft-based vertical profiles of $PM_{2.5}$ chemical components in 2016 winter for denormalizing the normalized vertical 290 profiles generated by the deep learning module are provided by Liu et al. (2020)'s study (Liu et al., 2020). The aircraft-based vertical profiles for retrieval independent verification were sampled in a flight experiment at an airport site in Shijiazhuang (37.54°N, 114.35°E). The flight time schedules (LT, local time) are detailed in Table 1. The tower-based vertical profiles at altitudes of 16 m, 102 m and 280 m were sampled at a 325-m meteorological tower located at the IAP, CAS in Beijing (39.98°N,116.38°E) during 10 days (27 and 30 December 2023; 2, 5, 9, 12, 15, 18, 24, and 27 January 2024). A flow 295 sampler with a flow rate of 42.8 L min$^{-1}$ and the 47-mm quartz filter membranes were utilized to collect $PM_{2.5}$ chemical component samples in the aircraft-based and tower-based sampling experiments. Furthermore, the 325 m tower-based vertical profiles on December 31 2018 were also collected from Lei et al. (2021)'s study (Lei et al., 2021).

**Table 1: Flight time schedules (LT, local time), corresponding surface temperature and relative humidity.**

| Date | Flight time | Sampling height (m) | Surface temperature (°C) | Surface relative humidity (%) |
|---|---|---|---|---|
| September 26 2024 | 19:10-21:10 | 2100 | 19.2-22.9 | 87.5-95 |
| October 10 2024 | 19:40-21:40 | 600 | 18.8-19.2 | 29-30 |
| December 11 2024 | 15:00-16:00 | 1200 | 4.3-4.9 | 31-34 |
| December 11 2024 | 16:00-17:00 | 1500 | 3.3-4.3 | 34-38 |

### 300 3 Results and discussion

### 3.1 Validation

### 3.1.1 Evaluation of the training and testing phase

The performance of the deep learning module within the retrieval framework, which was trained using a randomly partitioned multivariate dataset, is evaluated based on the discrepancies between simulations and observations for $NH_4^+$,



$SO_4^{2-}$, $NO_3^-$, OM and BC. Overall, the scatter distribution and fitted regression line closely align with the 1:1 line in both the training (Fig. 3a1-a5) and testing phases (Fig. 3b1-b5). The error distributions are concentrated around 0, with mean biases between $0.02 \pm 0.64$ µg m$^{-3}$ and $0.25 \pm 4.05$ µg m$^{-3}$ during the training phase (Fig. 3c1-c5) and between $0.03 \pm 0.81$ µg m$^{-3}$ and $0.20 \pm 5.82$ µg m$^{-3}$ during the testing phase (Fig. 3d1-d5), demonstrating strong consistency between observations and simulations. Specifically for the training process (Fig. 3a1-a5), the values of CORR and R$^2$ for the five PM$_{2.5}$ chemical

components range from 0.90 to 0.96 and from 0.78 to 0.93, respectively, indicating that the deep learning module accurately interprets the relationship between multivariate input features and the five PM$_{2.5}$ chemical components. The values of RMSE and MAE range from 0.64 to 4.08 µg m$^{-3}$ and from 0.46 to 2.76 µg m$^{-3}$, indicating a low discrepancy between simulations and observations. Compared to the training process, the values of CORR and R$^2$ during the testing process decrease to 0.84-0.91 and 0.64-0.82, respectively, and the values of RMSE and MAE increase to 0.81-5.82 µg m$^{-3}$ and 0.56-3.47 µg m$^{-3}$,

showing a slight underestimation for the five PM$_{2.5}$ chemical components (Fig. 3b1-b5). It is expected that the statistical results of the testing phase will be less robust than those of the training phase, as the deep learning module has not been trained on the data from the testing phase. Nevertheless, our statistical results from the testing phase exhibit similarities or even improvements compared to those reported in other studies that predicting PM$_{2.5}$ chemical component concentrations based on machine learning models (Lv et al., 2021; Lin et al., 2022; Araki et al., 2022; Liu et al., 2023), indicating that the

deep learning module demonstrates strong prediction capabilities.





**Figure 3:** Scatterplots of the simulations (µg m$^{-3}$) versus the observations (µg m$^{-3}$) with probability density (%) for NH$_4^+$, SO$_4^{2-}$, NO$_3^-$, OM and BC during the training process (a1-a5) and testing process (b1-b5). The dotted grey lines represent the 2:1, 1:1, and 1:2 lines, and the solid red line represents the fitted regression line. CORR represents the correlation coefficient, RMSE represents root mean square error, MAE represents mean absolute error, and R$^2$ represents determination coefficient. Probability distributions of bias (observations minus simulations, µg m$^{-3}$) for NH$_4^+$, SO$_4^{2-}$, NO$_3^-$, OM and BC during the training process (c1-c5) and testing process (d1-d5).

### 3.1.2 Comparison with ground-level observations

The retrieval framework was applied to retrieve the vertical profiles of NH$_4^+$, SO$_4^{2-}$, NO$_3^-$, OM and BC in a Beijing lidar site (39.98°N,116.38°E) over a training year (2021) and three non-training years (2017, 2018 and 2024). As illustrated in Fig. 4a1-a5, the weekly-smoothed variations in the retrieved surface concentrations of the five PM$_{2.5}$ chemical components demonstrate strong consistency with the observed surface concentrations for the training year, indicating that the retrieval framework adequately captures the temporal characteristics of these chemical components. The CORR values between the retrieved and observed concentrations range from 0.91 to 0.98, surpassing those of the deep learning module (Fig. 4a1-a5



and Fig. 3b1-b5), suggesting that the physics-constrained optimization effectively enhances the retrieval accuracy of chemical component concentrations. The RMSE values for $NH_4^+$ $SO_4^{2-}$ and $NO_3^-$ are marginally higher than those of the deep learning module (Fig. 4a1-a3 and Fig. 3b1-b3). Conversely, OM and BC exhibit a slight overestimation (Fig. 4a4, a5), which is attributed to inaccuracies in the upper boundaries of the scale factor during the physics-constrained optimization

process.

For the non-training years, the retrieved surface concentrations of a sum of five $PM_{2.5}$ chemical components are compared to the observed surface $PM_{2.5}$ concentrations, owing to the absence of long-term observations for individual chemical components. As shown in Fig. 4b-d, the weekly-smoothed variations in the retrieved surface $PM_{2.5}$ concentrations closely

align with the observed values in 2017, 2018 and 2024. The high values of surface $PM_{2.5}$ concentration observed in March-April and November of 2018 and 2024 are effectively captured by the retrieval framework. These results indicate that the retrieval framework exhibits robust temporal generalization capabilities, accurately interpreting the changes in concentrations of various chemical components across different periods. However, the retrieved concentrations show a slight overestimation relative to the observed values, potentially associated with the overestimation of carbonaceous aerosols as

reported in the training-year results. Future efforts should enhance retrieval accuracy by regulating the parameters involved in the physics-constrained optimization process.







**Figure 4: Weekly-smoothed variations in the retrieved and observed concentrations (µg m$^{-3}$) of NH$_4^+$ (a1), NO$_3^-$ (a2), SO$_4^{2-}$ (a3), OM (a4) and BC (a5) in 2021. (b) same as (a1-a5) but for PM$_{2.5}$ in 2017. (c) same as (a1-a5) but for PM$_{2.5}$ in 2018. (d) same as (a1-a5) but for PM$_{2.5}$ in 2024.**

The retrieval framework was also applied to retrieve the vertical profiles of the five PM$_{2.5}$ chemical components at 39 non-training BTH lidar sites over a short-term period of February 8-15$^{th}$, 2021, aiming to validate its spatial generalization capabilities. As shown in Fig. 5a, the retrieved and observed surface concentrations at 39 non-training sites exhibit comparable scatter distributions and closely aligned averages, indicating that the retrieval framework possesses reasonable spatial extensibilities, albeit with some overestimations in OM and BC. From a spatial perspective (Fig. 5b1-b5), non-





training BTH sites located closer to the Beijing lidar site exhibit higher CORR values, with the highest reaching 0.71 ($NH_4^+$),

0.56 ($NO_3^-$), 0.81 ($SO_4^{2-}$), 0.48 (OM) and 0.41 (BC). Conversely, the RMSE values are not affected by the distance from the

Beijing lidar site (Fig. 5c1-c5), with the lowest reaching 2.40 µg m⁻³ ($NH_4^+$), 4.65 µg m⁻³ ($NO_3^-$), 3.06 µg m⁻³ ($SO_4^{2-}$), 6.70

µg m⁻³ (OM) and 0.79 µg m⁻³ (BC). These findings suggest that the retrieval framework effectively retrieves PM$_{2.5}$ chemical

component concentrations at spatially varying lidar sites with minimal retrieval errors. However, it has limitations in

accurately describing short-term changes in carbonaceous aerosols at non-training sites.



**Figure 5: Scatter distribution of retrieved and observed surface mass concentration (µg m⁻³) of $NH_4^+$, $NO_3^-$, $SO_4^{2-}$, OM and BC at 39 non-training BTH lidar sites over a period of February 8-15th, 2021 (a). Spatial distribution of Pearson correlation coefficient (CORR) between retrieved and observed surface mass concentration of $NH_4^+$ (b1), $NO_3^-$ (b2), $SO_4^{2-}$ (b3), OM (b4) and BC (b5). (c1-c5) Same as (b1-b5) but for root mean square error (RMSE, µg m⁻³).**

**3.1.3 Verification of retrieved vertical profiles**

In addition to the spatiotemporal verification of surface-level mass concentrations, we conducted tower-based and aircraft-

based observational experiments to validate the retrieved vertical profiles of five PM$_{2.5}$ chemical components during the non-



training periods. From the surface level to ~200 m altitude, the retrieved and observed vertical profiles exhibit consistent vertical patterns with higher concentrations occurring at an altitude of ~120 m for $NH_4^+$, $NO_3^-$, $SO_4^{2-}$, OM and BC (Fig. 6a1, a2). Especially, the retrieved and observed mass concentrations of OM are comparable at ~120 m, with reported averages of 22.21 µg m$^{-3}$ and 24.82 µg m$^{-3}$, respectively. Additionally, the retrieved and observed proportions of $NH_4^+$, $NO_3^-$, $SO_4^{2-}$, OM and BC demonstrate significant consistency (Fig. 6b1, b2). Among these chemical components, $NO_3^-$ and OM represent the highest proportions, followed by $NH_4^+$ and $SO_4^{2-}$, while BC accounts for the lowest proportion. This proportional characteristic is evident in the retrieved and observed proportions at altitudes of 600 m and 1200 m (Fig. 6c1, c2). Due to the lack of $NH_4^+$ measurements at 1500 m and the absence of both $NH_4^+$ and $SO_4^{2-}$ measurements at 2100 m, the proportions at these altitudes are statistically obtained from other chemical components. The results indicate that the retrieved and observed proportions at altitudes of 1500 m and 2100m are consistent, although the proportion of $NO_3^-$ is somewhat overestimated. Overall, the tower-based and aircraft-based verifications indicate that the retrieval framework achieves high accuracy in retrieving the vertical profiles of the five $PM_{2.5}$ chemical components during the non-training period, demonstrating its robust generalization capabilities in generating high-precision vertical profiles from non-training datasets.





**Figure 6:** Vertical profiles (µg m$^{-3}$) of NH$_4^+$, NO$_3^-$, SO$_4^{2-}$, and OM from retrieval (a1) and tower-based observation (a2) for December 31, 2018. The line represents the daily average of the hourly vertical profiles, and the shaded area represents the standard deviation. Averaged proportions of NH$_4^+$, NO$_3^-$, SO$_4^{2-}$, OM, and BC from retrieval (b1) and tower-based observation (b2) during 10 days (December 27 and 30, 2023; January 2, 5, 9, 12, 15, 18, 24, and 27, 2024). (c1 and c2) Same as (b1 and b2) but for aircraft-based verification during 3 days (September 26, October 10, December 11, 2024).



### 3.2 Assessment of feature importance

The predictive performance of the deep learning module is intricately connected to the input features (Blum and Langley,
1997). Although the module incorporates the CNN and attention mechanism layer to mitigate issues related to feature
dimension, the impact of input features on the module predictions remains ambiguous, which impedes module
interpretability and restricts the capacity to enhance the module performance through effective feature selection. The SHAP
method is employed to quantify the relative contributions of 8 input features to the predictions of the five $PM_{2.5}$ chemical
components at various heights and to identify the impact of the input features on the decision-making processes of the deep
learning module. The coexistence of a high feature value with a positive SHAP value in a specific feature implies an
amplification of concentration prediction at elevated levels.

Figure 7a1-a5 depicts that the aerosol extinction coefficient at 532 nm (EXT), relative humidity (RH) and v-component wind
(VW) are the dominant input features for predicting the five $PM_{2.5}$ chemical components with an averaged relative
contribution of 14.43 %, 15.84 % and 16.77 %. These features determine the vertical structure, chemical and physical
processes, respectively. Specifically, EXT characterizes the vertical distribution of a total of the five $PM_{2.5}$ chemical
components and plays a crucial indicative role in vertical profile predictions (Tao et al., 2016). RH is a key driving factor in
aerosol hygroscopic growth, aqueous-phase chemical reactions, and heterogeneous reactions, significantly contributing to
the mass concentrations of varying chemical components as reported in numerous studies (Fang et al., 2019; Wang et al.,
2020; Gao et al., 2020; Liang et al., 2019). VW primarily affects latitudinal transboundary transport, which is a dynamic
forcing in the southwest-northeast transport channel of the BTH region (Yang et al., 2024). Notably, the relative contribution
of EXT decreases with height from the surface (50 m) to the free atmosphere (1900 m), while the relative contribution of
VW exhibits an opposite trend. The aerosol content in the upper atmosphere is relatively low, and the weakened lidar aerosol
signal is susceptible to interference from noise signals, restricting the indicative effect of EXT on chemical component
concentrations. Conversely, pollution transport in the upper atmosphere is less affected by interference from complex
underlying surfaces than near-surface transport (Wu et al., 2019b), amplifying the driving effect of high-altitude VW on
chemical component concentrations.





Figure 7: Relative contribution of 8 input features on predictive $NH_4^+$ (a1), $NO_3^-$ (a2), $SO_4^{2-}$ (a3), OM (a4) and BC (a5) at altitudes of 50 m, 766 m and 1900 m. SHAP values with feature values of 8 input features for predictive $NH_4^+$ (b1), $NO_3^-$ (b2), $SO_4^{2-}$ (b3), OM (b4) and BC (b5) at an altitude of 50 m. (c1-c5) Same as (b1-b5) but for an altitude of 766 m. (d1-d5) Same as (b1-b5) but for an altitude of 1900 m. F1: extinction coefficient at 532 nm, EXT; F2: Geopotential, GEOP; F3: Relative humidity, RH; F4: Specific humidity, SH; F5: Temperature, TEMP; F6: U-component wind, UW; F7: V-component wind; F8: Vertical velocity, VV.

Figure 7b1-d5 further determines the impact of the input features on the decision-making processes of the deep learning module. From Fig. 7b1-b5, the elevated levels of EXT, GEOP, and VW significantly enhance the concentration predictions of the five PM$_{2.5}$ chemical components in the near-surface layer (50 m), while high-level RH exert either positive or negative effects on predictions. High RH not only facilitates aqueous-phase and heterogeneous chemical reactions, positively contributing to predictions, but also promotes aerosol coalescence, leading to dry and wet deposition that negatively



contributes to predictions (Chen et al., 2020). The results in the middle of the boundary layer (766 m) are consistent with those observed in the near-surface layer (Fig. 7c1-c5). Particularly, the positive driving effect of lower VV values on predictions is more significant, with downward wind contributing positively to predictions, which is attributed to the fact that sinking airflows inhibit the dispersion of chemical components, thereby exacerbating aggregation and increasing concentration (Yang et al., 2022). The results in the free atmosphere (1900 m) align with those in the middle of the boundary layer (Fig. 7d1-d5). Notably, the influence of UW on predictions is more apparent, as the westerly wind positively contributes to the predictions, which is primarily due to the elevated emission sources located in the southwestern BTH region (Yang et al., 2024). Strong prevailing southwesterly winds at high altitudes enhance the regional transport of atmospheric pollutants, leading to an increase in concentration.

### 3.3 Application of the retrieval framework

The retrieval framework was applied to generate a long-term dataset of vertical profiles for $NH_4^+$, $NO_3^-$, $SO_4^{2-}$, OM and BC over six years (2017-2018, 2021-2024) at a Beijing lidar site. Figure 8 shows the averaged vertical profiles for the five $PM_{2.5}$ chemical components in spring (MAM) (Fig. 8a1), summer (JJA) (Fig. 8a2), autumn (SON) (Fig. 8a3) and winter (DJF) (Fig. 8a4) during the six years. OM mass concentrations are consistently the highest across all four seasons, followed by $NO_3^-$, while the mass concentrations of $NH_4^+$, $SO_4^{2-}$ and BC remain relatively low. The high proportions of OM and $NO_3^-$ in Chinese $PM_{2.5}$ pollution were frequently reported in recent studies (Zhang et al., 2024; Liu et al., 2022). Since the implementation of the Air Pollution Prevention and Control Action Plan during 2013-2017 and the Three-year Action Plan to Win the Blue-Sky Defense War during 2018-2020 in China, effective reductions in sulfur dioxide ($SO_2$) have gradually shifted the dominated chemical component of $PM_{2.5}$ pollution from $SO_4^{2-}$ to OM and $NO_3^-$ (Niu et al., 2022). Furthermore, the decreased $SO_4^{2-}$ mass concentrations have amplified the competitive effect of $NO_3^-$ on capturing $NH_3$ and $NH_4^+$ in the thermodynamic equilibrium process, increasing $NO_3^-$ mass concentrations (Geng et al., 2024). Notably, $NO_3^-$ mass concentration peaks at an altitude of ~310 m across all four seasons due to an enhanced formation potential resulting from gas-particle partitioning and heterogeneous reactions at elevated altitudes (Zhou et al., 2018). In comparison to the mass concentrations of the five $PM_{2.5}$ chemical components in MAM, SON and DJF, summertime mass concentrations are notably lower, which are attributed to reduced heating activities and enhanced wet deposition during summer periods (Liu et al., 2015; Ji et al., 2019). Moreover, the summertime vertical distributions of the five chemical components are relatively uniform, which may be attributed to the enhanced atmospheric vertical mixing effects induced by the unstable boundary layer (Roostaei et al., 2024).





**Figure 8: Vertical distribution of mass concentrations (µg m$^{-3}$) for NH$_4^+$, NO$_3^-$, SO$_4^{2-}$, OM and BC in spring (MAM, a1), summer (JJA, a2), autumn (SON, a3) and winter (DJF, a4) over six years (2017-2018, 2021-2024). Averaged vertical profiles of mass concentrations (µg m$^{-3}$) for NH$_4^+$, NO$_3^-$, SO$_4^{2-}$, OM and BC from 2017 to 2018 (b1), from 2021 to 2022 (b2), and from 2023 to 2024 (b3). Annual change rates (µg m$^{-3}$ a$^{-1}$) of mass concentrations for NH$_4^+$, NO$_3^-$, SO$_4^{2-}$, OM and BC at various altitudes from 2021 to 2024 (c).**

Figure 8 also shows the averaged vertical profiles during 2017-2018 (Fig. 8b1), 2021-2022 (Fig. 8b2) and 2023-2024 (Fig. 8b3) and the annual change rate during 2021-2024 (Fig. 8c). The implementation of clean air policies during 2017-2018 resulted in mass concentrations of NH$_4^+$, SO$_4^{2-}$ and BC remaining below 11 µg m$^{-3}$ (Fig. 8b1). However, the mass concentrations of NO$_3^-$ exceeded 17 µg m$^{-3}$ at altitudes of below 100 m and ~310 m, and the mass concentrations of OM exceeded 36 µg m$^{-3}$ at altitudes of below 200 m due to the nonlinear response to emission reduction (Li et al., 2021). Compared to 2017-2018, the mass concentrations of NH$_4^+$, NO$_3^-$, SO$_4^{2-}$, OM and BC decreased significantly during 2021-2022 with average reductions of 28.29 %, 22.70 %, 29.55 %, 56.78 % and 57.43 % from 50 m to 3000 m, respectively (Fig.




8b2), which is induced by the continued implementation of clean air policies and reduced emissions associated with the
COVID-19 pandemic control in China (Kang et al., 2020). During 2023-2024, the mass concentrations of $NH_4^+$, $NO_3^-$, $SO_4^{2-}$,
OM and BC exhibited slight reductions compared to the 2021-2022 levels, with average decreases of 6.57 %, 7.71 %,
14.22 %, 0.12 % and 5.58 % from 50 m to 3000 m, respectively (Fig. 8b3), which is related to the offsetting effect of
enhanced human activities following the relaxation of the COVID-19 pandemic lockdowns on the implementation of clean
air policies (Song et al., 2025). From 2021 to 2024, except for an altitude of ~200 m, the reduction rates of the five chemical
components are largely consistent across varying altitudes within the boundary layer (below ~1900 m), with the highest
reduction rate of 0.17-0.82 µg m$^{-3}$ a$^{-1}$ occurring at an altitude of ~300 m (Fig. 8c). However, OM exhibited a significant
increase rate of 1.84 µg m$^{-3}$ a$^{-1}$ at an altitude of ~200 m, which may be related to the low sensitivity of high-altitude organic
aerosols to emission controls (Zhao et al., 2017). Future clean air policies should prioritize strengthening control measures
for OM and $NO_3^-$ within the lower and middle parts of the atmospheric boundary layer.

**4 Conclusions**

This study proposes a novel lidar-based retrieval framework for obtaining the vertical profiles of five PM$_{2.5}$ chemical
components ($NH_4^+$, $SO_4^{2-}$, $NO_3^-$, OM and BC) for the first time. A long-term multivariate dataset was utilized to train a
complex deep-learning module in the retrieval framework, thus interpreting the nonlinear relationship among lidar
parameters, meteorological parameters and PM$_{2.5}$ chemical components. A physics-constrained optimization module was
integrated into the retrieval framework, enhancing the generalization capabilities of predicting vertical profiles across diverse
spatiotemporal scenarios.

In situ surface observations of hourly mass concentrations of PM$_{2.5}$ and its five chemical components over a training year
and three non-training years were used to validate the accuracy of the retrieval framework in interpreting temporal variations.
The results showed that the Pearson correlation coefficient values between the retrieved and observed concentrations ranged
from 0.91 to 0.98 during the training year, and the variations in the retrieved surface PM$_{2.5}$ mass concentrations closely
aligned with the observations during the non-training year, indicating the robust capabilities of temporal prediction and
generalization in the retrieval framework. Then the retrieval framework was applied to obtain mass concentrations of five
PM$_{2.5}$ chemical components at 39 non-training sites, which exhibited patterns consistent with the corresponding observations
for $NH_4^+$, $SO_4^{2-}$ and $NO_3^-$. However, limitations remained in accurately capturing short-term temporal variations in OM and
BC. Tower-based and aircraft-based field campaigns at altitudes ranging from surface to 2100 m were conducted to validate
the accuracy of the retrieved vertical profiles of $NH_4^+$, $SO_4^{2-}$, $NO_3^-$, OM and BC. The tower-based and aircraft-based
verifications indicate that the retrieved and observed vertical profiles of these components exhibited consistent patterns in
mass concentrations and proportions, demonstrating the robust capabilities of the retrieval framework in obtaining high-
precision vertical profiles from non-training datasets.

Subsequently, SHapley Additive exPlanations (SHAP), an explainable technology, is integrated into the deep learning module to quantify the impact of multivariate input features on the retrieval of $PM_{2.5}$ chemical components. The results showed that the aerosol extinction coefficient at 532 nm, relative humidity and v-component wind are the dominant input

features for predicting the five $PM_{2.5}$ chemical components with an averaged relative contribution of 14.43 %, 15.84 % and 16.77 %. The driving effect of the input features on the decision-making processes of the deep learning module was also determined by SHAP values.

Finally, we applied this framework to generate a long-term dataset of vertical profiles for $NH_4^+$, $SO_4^{2-}$, $NO_3^-$, OM and BC

over six years (2017-2018, 2021-2024). From this dataset, we found that OM mass concentrations are consistently the highest across all four seasons, followed by $NO_3^-$, while the mass concentrations of $NH_4^+$, $SO_4^{2-}$ and BC remain relatively low. From 2021 to 2024, except for an altitude of ~200 m, the reduction rates of the five chemical components are largely consistent across varying altitudes within the boundary layer (below ~1900 m), with the highest reduction rate of 0.17-0.82 $\mu g\ m^{-3}\ a^{-1}$ occurring at an altitude of ~300 m. However, OM exhibited a significant increase rate of 1.84 $\mu g\ m^{-3}\ a^{-1}$ at an

altitude of ~200 m. Future clean air policies should prioritize strengthening control measures for OM and $NO_3^-$ within the lower and middle parts of the atmospheric boundary layer. Our new retrieval framework offers a novel approach to acquiring vertical profiles of $PM_{2.5}$ chemical components. Future efforts should aim to mitigate the overestimation of carbonaceous aerosols by regulating the parameters involved in the physics-constrained optimization process.

**Data availability**

All data in this manuscript are freely available upon request through the corresponding author (tingyang@mail.iap.ac.cn).

**Author contributions**

HL developed the retrieval framework, carried out the analysis and verification, as well as wrote this paper. TY provided scientific guidance and wrote this paper. TY and YS provided various measurement data. ZW did overall supervision. All authors reviewed and revised this paper.

**Competing interests**

The authors declare that they have no conflict of interest.



**Acknowledgements**

Ting Yang would like to express gratitude towards the Program of the Youth Innovation Promotion Association (CAS). We thank for the technical support of the National large Scientific and Technological Infrastructure "Earth System Numerical Simulation Facility" (https://cstr.cn/31134.02. EL), and the data support of the China National Environmental Monitoring Center.

**Financial support**

This work was supported by National Natural Science Foundation of China (NSFC) Excellent Young Scientists Fund (grant no. 42422506), National Key Research and Development Program of China (grant no. 2023YFC3705801), and the National Natural Science Foundation of China (grant no. 42275122).

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
