# Peer review of "A Physics-Constrained Deep-Learning Framework based on Long-Term Remote-Sensing Data for Retrieving Vertical Distribution of PM2.5 Chemical Components"

_EGUsphere, 2025_

## Referee Comment (RC2)

**Review of "A Physics-Constrained Deep-Learning Framework for Retrieving Vertical Distribution of PM2.5 Chemical Components" (AMT)**

This manuscript presents a novel lidar-based retrieval framework that integrates deep learning with physics-constrained optimization to estimate vertical mass concentration profiles of five $PM_{2.5}$ chemical components ($SO_4^{2-}$, $NO_3^-$, $NH_4^+$, OM, BC). The topic is scientifically important: retrieving aerosol composition profiles from lidar would significantly advance air quality monitoring, chemical transport modeling, and source apportionment. The combination of physics constraints and deep learning is innovative and promising.

However, while the conceptual idea is strong, the manuscript lacks clarity in describing the model framework and provides insufficient evidence that the approach accurately captures the physics of vertical aerosol composition or generalizes across seasons, sites, and aerosol regimes. Significant issues in methodology, validation, and presentation hinder the scientific interpretation of the results. I therefore recommend **major revision**.

**Major Concerns**

**1. Overall framing, workflow clarity, and Figure 1**

- **Figure 1 is difficult to interpret**: inputs/outputs are not clearly labeled, colored boxes lack explanation, and several acronyms are undefined. The figure should be redesigned as a **clear block-flow diagram** that lists:
    - all inputs (with units, vertical resolution, and dimensionality),
    - each module's output,
    - loss functions used,
    - data flow direction and optimization loops.
- **Reorder sections** so the Data section precedes the Model description. Readers must understand what data the model consumes before interpreting architectural choices.

**2. Ambiguity in algorithm description**

- Section 2.1.1 is **confusing and lacks foundational background**, making the workflow difficult to follow without jumping back and forth.
- It is unclear **what the deep-learning model predicts per vertical level**. Please explicitly specify:
    - whether the model outputs component concentrations, component fractions, categorical flags, or something else,
    - the *exact dimensionality* (e.g., levels × 5 components).
- Clearly define the **target variables** and how they are constructed.
- Provide detailed descriptions of the **multi-objective optimization**, including:
    - inputs and outputs,
    - spatial/temporal/vertical resolution,
    - how physics constraints are incorporated mathematically.
- The purpose of using mentioned components/models.

- The rationale for using a **two-step prediction process** (component "flags" followed by concentrations) rather than a single multi-output network is not explained. The manuscript would benefit from an experimental justification or comparison.

**3. Temporal and spatial data splits**

- The current **random 80/20 split** is not appropriate for meteorological/aerosol time series due to temporal autocorrelation, which risks information leakage.
- Consider implementing:
  - **temporal holdouts** (e.g., full seasons),
  - **spatially independent test sites**,
  - **blocked k-fold cross-validation** preserving temporal/spatial independence.
- The manuscript evaluates an *independent* dataset only in the Results section, but this dataset should be partially used for the validation/testing framework.
- The reported error statistics for the independent dataset are **not clearly presented and differ considerably** from training results. For a well-generalized model, **validation and independent-test errors should be similar**; their discrepancies raise concerns about generalization and physical consistency.
- Surface-only scatterplots from the training year are insufficient to establish model validity, especially given that the model's primary output is a **vertical distribution**.

**4. Weak vertical-profile validation**

The manuscript focuses on retrieving vertical composition profiles but presents **minimal validation** of these profiles.

I strongly recommend including:

- Direct comparisons with **aircraft or in situ vertical measurements**, using metrics such as bias, RMSE, MAE, percent error, and correlation *at each altitude bin*.
- **Case studies** across representative aerosol regimes (smoke, dust, pollution, background).
- Aggregated statistics by:
  - altitude,
  - site,
  - aerosol type,
  - season.

If vertical observational data are limited, the manuscript should **explicitly quantify these limitations** while still presenting as much vertical validation as possible.

**5. Heterogeneous site performance**

- Figure 5b shows substantial site-to-site variability: some sites have nearly zero correlation, while the best site reaches ~0.6.
- Please investigate and report potential causes, such as:
  - aerosol-type mismatch,

- o representativeness of training data,
- o site-specific meteorology or emissions,
- o instrument characteristics.
- Consider:
  - o a map showing training vs. test sites,
  - o per-site metrics (MAE, RMSE, bias, percent error, N),
  - o problematic site scatterplots or boxplots to illustrate error spread.

**6. Lack of uncertainty quantification**

Given the physics-constrained framing, the model should also provide **uncertainty estimates**, or at minimum a discussion of uncertainty propagation. Possible approaches include: Ensemble modeling, Monte Carlo dropout, error propagation from lidar extinction + physics constraints. Uncertainty bounds would greatly strengthen confidence in profile retrievals.
* * *
**Minor Comments and Suggestions**

1. If possible, include an **ablation study** comparing architectures (CNN, BiLSTM, CNN+BiLSTM, transformer) to justify the chosen hybrid design.
2. Clarify the meaning of "estimated and observed extinction coefficients" (line 192). Does "estimated" refer to IMPROVE-derived extinction?
3. Define **all acronyms** at first use; ensure figure captions are self-contained.
4. The manuscript describes z-score normalization but not how **denormalization** is performed. Why use aircraft-based measurements for denormalization instead of lidar-derived extinction? Explain and quantify the sensitivity.
5. The scaling procedure using the ratio of in situ to aircraft $PM_{2.5}$ ("initially scaled…") is ambiguous. Provide a **clear mathematical expression** and discuss whether this introduces bias.
6. The description of the **attention layer** lacks physical interpretation. Is attention purely data-driven, or does physics guide attention weights? If physics influences attention, show how.
7. Figure 1 needs explicit legends for color boxes/arrows and clear annotation of all inputs and outputs.
8. Figure 5a does not effectively show differences between datasets. Consider:
   - o scatterplots colored by site with standard deviations,
   - o an additional plot showing error distribution histograms for each of the five components.
9. Include full **training hyperparameters**: batch size, learning rate, optimizer, epochs, early stopping criteria, normalization statistics.

---

## Author Comment (AC1)

**Authors' responses to Referees' comments**

**Journal:** Atmospheric Measurement Techniques

**Manuscript Number:** egusphere-2025-4237

**Title:** A Physics-Constrained Deep-Learning Framework based on Long-Term Remote-Sensing Data for Retrieving Vertical Distribution of PM$_{2.5}$ Chemical Components

**Authors:** Hongyi Li, Ting Yang, et al.

Note:

Comment (12-point black italicized font).

Reply (indented, 12-point blue normal font).

"Revised text as it appears in the text (in quotes, 12-point blue italicized font)".
* * *
**Anonymous Referee #1**

*1 General comments:*

*"A Physics-Constrained Deep-Learning Framework based on Long-Term Remote-Sensing Data for Retrieving Vertical Distribution of PM$_{2.5}$ Chemical Components" by Li et al., proposes a novel method to retrieve concentrations of major aerosol components (sulfate, nitrate, ammonium, organic matter, and black carbon) from ground-based lidar extinction data combined with ERA5 meteorological reanalysis. The authors validate their retrieved concentrations against surface, airborne, and tower observations in the greater Beijing region, reporting generally high correlations.* ***Given the importance of aerosols in Earth's radiative balance and air quality, developing methods that leverage lidar's high vertical resolution to determine constituent species concentrations is a valuable endeavor.*** *However, the manuscript in its current form requires significant revisions to adequately describe the methodology and contextualize the results.*

**Authors' response:**

We sincerely thank the Reviewer for the thoughtful assessment of our manuscript and for recognizing the potential value of our work. In response to the Reviewer's comments, we have undertaken a comprehensive revision of the manuscript.

*2 Major Comments:*

*1) A fundamental issue is how $PM_{2.5}$ concentrations are distinguished from larger particles using lidar extinction data. The lidar dataset presumably provides total aerosol extinction from particles of all sizes, yet the work presented here centers only on $PM_{2.5}$. The authors provide no explanation of how contributions from larger particles ($PM_{10}$, coarse mode, etc.) are excluded from the extinction signal (if they are). This represents a potentially significant source of error, particularly during dust events when coarse particles may dominate extinction.*

**Authors' response:**

We would like to thank the reviewer for the insightful suggestion. We fully concur that the identification of $PM_{2.5}$ from the total aerosol extinction signal influenced by coarse-mode particles presents a potential source of error. We tackled this issue by employing end-to-end machine learning training that utilizes long-term lidar signals and observations of $PM_{2.5}$ chemical compositions. The detailed explanations are as follows:

a. **The total aerosol extinction from particles of all sizes can serve as a direct indicator for retrieving $PM_{2.5}$.** Earlier studies have revealed a strong correlation between $PM_{2.5}$ and total light attenuation from particles of all sizes, such as aerosol optical depth (AOD) (van Donkelaar et al., 2010; Zhang et al., 2009) and aerosol extinction coefficient (EXT) (Lowenthal and Kumar, 2016; Tao et al. 2012). Therefore, the total light attenuation has been widely served as a crucial indicator for directly predicting $PM_{2.5}$ concentrations in machine learning models (Table 1).

**Table 1**. Literature review of retrieving $PM_{2.5}$ from the total light attenuation based on machine learning algorithms.

| Target feature | Optical input feature | Machine learning algorithm | Citation |
|---|---|---|---|
| Ground-level $PM_{2.5}$ | AOD at 550nm | CNN | Park et al., 2020 |
| Ground-level $PM_{2.5}$ | TOA reflectance at 460nm, 640nm and 2300nm | Geoi-LSTM | Wang et al., 2021 |
| Ground-level $PM_{2.5}$ | AOD at 470nm | XGBoost | Gutiérrez-Avila et al., 2022 |

| PM$_{2.5}$ vertical profile | AOD at 532nm | ET (best) | Chen et al., 2022 |
| PM$_{2.5}$ vertical profile | EXT at 580nm and 590nm | CNN-BiLSTM (best) | Yi et al., 2025 |
| PM$_{2.5}$ vertical profile | EXT at 532nm | CNN-BiLSTM | Wang et al., 2025 |

(CNN: Convolutional neural network; TOA: Top of atmosphere; Geoi-LSTM: Geo-intelligent Long Short-Term Memory; XGBoost: eXtreme Gradient Boosting; CALIOP: Cloud-Aerosol Lidar with Orthogonal Polarization; ET: Extra Trees; BiLSTM: Bidirectional LSTM)

b. **Machine learning methods can directly identify fine-mode particles from lidar extinction signals through end-to-end learning**. The retrieval framework in our work establishes a nonlinear mapping relationship between the extinction coefficient at 532nm and the chemical compositions of PM$_{2.5}$ to identify fine-mode particles from lidar extinction data. This end-to-end learning prevents the filtering out of lidar extinction signals induced by larger particles (such as PM$_{10}$), since the learning process specifically captures the signal patterns associated with variations in the chemical compositions of PM$_{2.5}$, which were the data fed into the learning process.

c. **Meteorological reanalysis data is used as auxiliary training data to strengthen the EXT-PM$_{2.5}$ relationship**. As the reviewer noted, coarse particles may predominate in extinction during dust events, making the identification of fine particles based solely on the extinction coefficient inaccurate. Following the previous studies (Lee et al., 2011; Xie et al., 2015), we integrated meteorological conditions (such as temperature, relative humidity, wind conditions and vertical velocity) into the machine-learning training process to better capture the spatiotemporal variations in the EXT-PM$_{2.5}$ relationship.

d. **Long-term training datasets allow machine learning models to learn the EXT-PM$_{2.5}$ relationship across various aerosol mode scenarios**. We utilized a 18-month dataset to establish a nonlinear mapping relationship between lidar signals and PM$_{2.5}$ chemical compositions. The training dataset encompasses various seasons and multiple aerosol mode scenarios, including strong dust events reported in Beijing during March 2021 (Gui et al., 2022). This diversity ensures that our machine learning model encounters a range of aerosol mode distributions and learns to make accurate predictions in the presence of coarse particles.

**Reference**

Chen, B., Song, Z., Pan, F., et al.: Obtaining vertical distribution of $PM_{2.5}$ from CALIOP data and machine learning algorithms, Sci. Total Environ., 805, 150338, https://doi.org/10.1016/j.scitotenv.2021.150338, 2022.

Gui, K., Yao, W., Che, H., et al.: Record-breaking dust loading during two mega dust storm events over northern China in March 2021: aerosol optical and radiative properties and meteorological drivers, Atmos. Chem. Phys., 22, 7905-7932, https://doi.org/10.5194/acp-22-7905-2022, 2022.

Gutiérrez-Avila, I., Arfer, K.B., Carrión, D. et al.: Prediction of daily mean and one-hour maximum $PM_{2.5}$ concentrations and applications in Central Mexico using satellite-based machine-learning models, J. Expo. Sci. Environ. Epidemiol., 32, 917-925, https://doi.org/10.1038/s41370-022-00471-4, 2022.

Lee, H. J., Liu, Y., Coull, B. A., Schwartz, J., and Koutrakis, P.: A novel calibration approach of MODIS AOD data to predict $PM_{2.5}$ concentrations, Atmos. Chem. Phys., 11, 7991-8002, https://doi.org/10.5194/acp-11-7991-2011, 2011.

Lowenthal, D. H., & Kumar, N.: Evaluation of the IMPROVE Equation for estimating aerosol light extinction, J. Air Waste Manage., 66(7), 726-737, https://doi.org/10.1080/10962247.2016.1178187, 2016.

Tao, J., Cao, JJ., Zhang, RJ. et al.: Reconstructed light extinction coefficients using chemical compositions of $PM_{2.5}$ in winter in Urban Guangzhou, China, Adv. Atmos. Sci. 29, 359-368, https://doi.org/10.1007/s00376-011-1045-0, 2012.

van Donkelaar, A., Martin, R., Brauer, M., et al.: Global estimates of ambient fine particulate matter concentrations from satellite-based aerosol optical depth: development and application, Environ. Health Perspect., 118(6), 847-855 https://doi.org/10.1289/ehp.0901623, 2010.

Wang, B., Yuan, Q., Yang, Q., et al.: Estimate hourly $PM_{2.5}$ concentrations from Himawari-8 TOA reflectance directly using geo-intelligent long short-term memory network, Environ. Pollut., 271, 116327, https://doi.org/10.1016/j.envpol.2020.116327, 2021.

Xie, Y., Wang, Y., Zhang, K., et al.: Daily Estimation of Ground-Level $PM_{2.5}$ Concentrations over Beijing Using 3 km Resolution MODIS AOD, Environ. Sci. Technol., 49, 20, 12280-12288, https://doi.org/10.1021/acs.est.5b01413, 2015.

Yi, Z., Xiang, Y., Yun L., et al.: Deep learning-driven reconstruction of $PM_{2.5}$ vertical profiles: A fusion of lidar and tower data, J. Clean Prod., 502, 145397, https://doi.org/10.1016/j.jclepro.2025.145397,

2025.

Zhang, H., Hoff, R. M., & Engel-Cox, J. A.: The Relation between Moderate Resolution Imaging Spectroradiometer (MODIS) Aerosol Optical Depth and PM$_{2.5}$ over the United States: A Geographical Comparison by U.S. Environmental Protection Agency Regions, J. Air Waste Manage., 59(11), 1358-1369. https://doi.org/10.3155/1047-3289.59.11.1358, 2009.

*2) The data processing section, especially the lidar data processing, lacks essential technical details. Key missing information includes lidar instrument specifications, data quality control procedures (e.g, cloud screening), lidar extinction retrieval algorithms/methods, and the methods for reconciling the different vertical resolutions between the lidar data (6m) and the meteorological inputs.*

**Authors' response:**

We sincerely apologize for the omission of technical details related to lidar specification parameters and data processing in the original manuscript. In response to the reviewers' feedback, we have revised the original manuscript and supplemented the content in the supplementary materials. ***Section 2.1*** was exchanged with ***Section 2.2*** for better description. The specific additions are outlined below.

Section 2.1.1: "*The $\sigma_{bsc,532}$ data for deep learning module training and PM$_{2.5}$ chemical component retrieving is obtained from a ground-based dual-wavelength polarization Mie lidar at the Institute of Atmospheric Physics (IAP), Chinese Academy of Sciences (CAS), Beijing (39.98°N,116.38°E). This Mie lidar has consistently detected optical signals since 2017, offering a temporal resolution of 15 minutes and a vertical resolution of 6 m. The lidar specification parameters and data preprocessing are detailed in Text S1 (and Table S1) and Text S2 of the supplement, respectively. The $\sigma_{bsc,532}$ data from February 8-15$^{th}$, 2021 at 23 lidar sites in the North China Plain (NCP), provided by the China National Environmental Monitoring Center (CNEMC), were utilized to assess the spatial generalization ability. The multi-site data offers a temporal resolution of 5-20 minutes and a vertical resolution of 7.5 m. To generate an hourly resolution lidar dataset, minute-level data were resampled using a simple*

*averaging method. Specifically, the arithmetic mean was calculated from all valid minute-level data points within each non-overlapping one-hour window aligned to the start of each hour (e.g., from 00:00 to 00:59)."*

: "*...The grid cells of EAC4 and ERA5 that contain the lidar sites were extracted using the k-nearest neighbor search method based on longitude and latitude data (Friedman et al., 1977). The lidar data and the reanalysis data were interpolated onto a preset vertical grid with a height range of 50 m to 3 km using linear interpolation. The preset height information is presented in Text S2 of the supplement.*"

Supplement, Text S1, Lidar instrument specifications: "*As shown in Table S1, the laser emission at wavelengths of 532 nm and 1064 nm relies on a Nd:YAG laser with a second harmonic generator and is corrected by a beam expander before emission. The emitted laser energies at 532 nm and 1064 nm are 30 mJ and 20 mJ, respectively. The laser pulse repetition frequency can reach up to 20 Hz and is set to 10 Hz in practice. The scattered light is collected by a Schmidt-Cassegrain telescope with a diameter of 20 cm and then is collimated and corrected toward a dichroic mirror to separate the received lidar signals at 532 nm and 1064 nm. The lidar signal at 532 nm is separated into horizontal and vertical polarization components and is measured by a photomultiplier tube. The lidar signal at 1064 nm is directly detected by an avalanche photodiode. Finally, the detected lidar signals are recorded by a digital oscilloscope and then are transferred to a computer for data storage.*"

**Table S1**. The main specification parameters of dual-wavelength polarization Mie Lidar.

| Parameter categories | | Description |
|---|---|---|
| Laser type | | Flashlamp pumped Nd:YAG |
| Laser pulse energy | 532 nm | 30 mJ/pulse |
| | 1064 nm | 20 mJ/pulse |
| Pulse Repetition Frequency | | ≤ 20 Hz, 10 Hz used in this work |
| Telescope Type | | Schmidt Cassegrain |
| Telescope diameter | | 20 cm |
| Field of view | | 1 mrad |
| Detector type | 532 nm | Photomultiplier tube (PMT) |
| | 1064 nm | Avalanche photodiode (APD) |
| Data acquisition system | | Digital oscilloscope |

: "*A comprehensive data quality control procedure was implemented on the original lidar signals to mitigate issues raised by electrical signal errors and signal offsets caused by background radiation. First, background noise was removed by subtracting the average value of signals within the altitudes of 3-9 km from the original lidar signal. Second, the lidar signal was range-corrected by multiplying by the square of the altitude and corrected for the geometric overlap effect using an empirically determined function derived from lidar profiles under well-mixed atmospheric conditions. Third, a cloud-screening algorithm was applied to identify and remove profiles contaminated by clouds. The algorithm operates by first calculating the vertical gradient of the range-corrected signal. It then identifies potential cloud bases as regions where this gradient exceeds a primary threshold of 4 $\times 10^{-8}$ for at least 3 consecutive resolution layers. For each candidate cloud layer, the algorithm determines the cloud top and then validates the layer by checking if the maximum signal within it surpasses a secondary threshold of 5 $\times 10^{-6}$. Profiles containing such validated cloud layers were entirely excluded from the subsequent aerosol analysis. Finally, the extinction coefficient at a wavelength of 532 nm was retrieved based on Fernald algorithm (Fernald, 1984).*

*To facilitate data fusion and comply with the input requirements of the machine learning model, all data were vertically re-sampled onto a standardized set of preset height levels ranging from 50 m to 3 km. The high-resolution lidar data and the low-resolution global reanalysis data were interpolated onto this uniform vertical grid using linear interpolation. The preset height grid with logarithmic intervals can be determined by Eq. S1-S2. Logarithmic interval amplifies vertical resolution within the planetary boundary layer, where fine-mode particles and their chemical components are typically most concentrated (Yang et al., 2024).*

$$h_i = 10^{log_{10}(Z_{min})+(i-1)\times \Delta Z}, i = 1, 2, ..., n \qquad (S1)$$

$$\Delta Z = \frac{log_{10}(Z_{max}) - log_{10}(Z_{min})}{n-1} \qquad (S2)$$

*Where $h_i$ is the height at $i^{th}$ vertical layer, $Z_{min}$ is the minimum height, $\Delta Z$ is the logarithmic interval, $Z_{max}$ is the maximum height, and $n$ is the total number of*

*vertical layers.*"

**Reference**

Fernald, F. G.: Analysis of atmospheric lidar observations: some comments, Appl. Opt., 23, 652-653, https://doi.org/10.1364/AO.23.000652, 1984.

Yang, T., Li, H., Xu, W., Song, Y., Xu, L., Wang, H., Wang, F., Sun, Y., Wang, Z., and Fu, P.: Strong Impacts of Regional Atmospheric Transport on the Vertical Distribution of Aerosol Ammonium over Beijing, Environ. Sci. Technol. Lett., 11, 29-34, https://doi.org/10.1021/acs.estlett.3c00791, 2024.

*3) The validation dataset is insufficient to support the broad conclusions presented. Aircraft validation comprises only four flights (limited to three different calendar months), while tower measurements span just 11 days across two time periods. This is of particular importance because the retrieved aerosol concentrations appear to show similar vertical distributions across different seasons. The limited validation prevents assessment of whether this method captures realistic atmospheric processes or simply learns scaling relationships under specific (mostly wintertime) meteorological conditions.*

**Authors' response:**

We fully agree that robust and comprehensive validation is essential to support the generalizability of our retrieval framework in spatiotemporally varying scenarios. In this work, we designed a multi-faceted validation strategy to thoroughly validate the model's generalizability from different perspectives by using the non-training dataset. The non-training dataset contains spatiotemporal information that the machine learning model has never learned, enabling a validation of whether this method captures realistic atmospheric processes. The validation strategy is outlined as follows.

**a**. The independent validation/testing sets, which were not used in deep learning, are employed to evaluate whether the nonlinear mapping relationship established by the model is reliable. As presented in ***Fig. 3*** **of** ***Section 3.1.1***, the results demonstrated high predictive accuracy on the independent validation/testing sets, with five $PM_{2.5}$ chemical species showing high agreement (R $\geq$ 0.69, RMSE $\leq$ 8.87 μg m$^{-3}$) with ground

observations, indicating that our model can learn complex and nonlinear relationships rather than memorizing the training data.

**b**. We utilized an independent ground observation set of $PM_{2.5}$ concentrations from three years not included in model training to validate the temporal generalization of our method. The sum of our retrieved $PM_{2.5}$ chemical species concentrations was compared against these independent measurements. As presented in ***Fig. 4b, c and d* of *Section 3.1.2***, the results indicate that our method accurately characterizes the changes in mass concentrations of various $PM_{2.5}$ chemical components across all seasons and under diverse meteorological conditions, not just the wintertime conditions.

**c**. We utilized a ground observation set of chemical component concentrations from 23 independent sites across the North China Plain (NCP) to validate the spatial generalization of our method. As presented in ***Fig. 5* of *Section 3.1.2***, comparisons between our retrieved surface concentrations and measurements from these untrained sites showed moderate agreement, indicating that our method is not site-specific but possesses robust predictive power on a regional scale.

**d**. We acknowledge the well-noted challenge of acquiring high-frequency vertical profile observation data. The validation results based on 4 aircraft campaigns and 11-day tower-based measurements showed that our retrieved vertical profiles of chemical components showed encouraging consistency with observations in both shape and magnitude. Most importantly, these vertical validation cases must be interpreted within the context of the replies of **Major Comments #3b, c**. The strong performance in spatiotemporal generalization provides a foundational credibility that our method can capture realistic relationships between extinction coefficient and chemical species (**Major Comments #3b, c**). The vertical validation then confirms that these relationships correctly translate into accurate vertical structures (**Major Comments #3d**).

We acknowledge that acquiring more observations from tower-based and aircraft-based campaigns is essential for adequately strengthening the validation of our method. We openly discuss the limitations regarding the sample size of vertical validation data

in ***Section 3.4 Limitations and uncertainties.*** Besides, we emphasize the significant role of temporal generalization validation in addressing concerns about seasonal representativeness in ***Section 2.2.4***.

Section 2.2.4, Line 330-332: "*… (1) We compare the retrieved mass concentrations with the observed values at the surface level during a training year (2021) and three non-training years (2017, 2018 and 2024) to validate the temporal generalization in all seasons and under diverse meteorological conditions…*"

Section 3.4: "*The deep learning module in our retrieval framework can establish a powerful mapping between optical and meteorological features and $PM_{2.5}$ chemical species, and physics-based explicit constraints can enhance the reliability and expandability of the mapping relationships. However, several limitations and sources of uncertainty remain and should be acknowledged when interpreting the results and extending the framework to broader applications.*

*First, the spatial scope of the training data is predominantly restricted to the NCP region. Expanding the retrieval framework with data from more diverse geographical locations is necessary to improve its global transferability. Second, the current retrieval framework primarily relies on extinction coefficients at a wavelength of 532 nm, exhibiting dependence on specific lidar instruments. Future retrieval framework should focus on integrating diverse optical features from additional wavelengths to enhancing adaptability and transferability. Third, the auxiliary input data used in both the deep learning module and the physics-constrained optimization are obtained from global reanalysis products, which may not fully capture local atmospheric conditions at specific observational sites, thereby introducing representativeness errors into the retrievals. Acquiring the vertical observational data for these auxiliary features can effectively mitigate the uncertainty induced by the input data. Fourth, the IMPROVE equation applied as an external physical constraint may introduce additional uncertainty into the retrievals due to its systematic estimation biases (Lowenthal and Kumar, 2016). Moreover, since the IMPROVE equation was applied as an external*

*physical constraint to optimize the retrievals of PM$_{2.5}$ chemical components, the machine learning model itself was not intrinsically constrained by physical principles during its training. Future work could incorporate an internal physical constraint into the machine learning model to improve its physical interpretability by formulating a hybrid loss function for training that combines the traditional data-fitting term with a physical term. Finally, long-term acquisition of independent vertical profiling data from both tower-based and aircraft-based campaigns is essential for a comprehensive assessment of the robustness of the vertical retrievals with respect to varying sites, aerosol types, and seasons.*"

*4) The manuscript would benefit from some extensive editing to improve its readability. Sentences are overly dense, and the model development section would be difficult for most readers to follow. The excessive number of figures (~75 figure/subplots) dilutes the presentation of key results.*

**Authors' response:**

We sincerely thank the reviewer for this critical feedback regarding the readability and presentation of the manuscript. In response to the reviewer's suggestions, we have undertaken a comprehensive revision of the manuscript, including the entire text, all figures, and their captions. Several detailed supporting subfigures have been moved to the supplement materials to maintain a clean and focused flow in the main text while still providing all necessary data for interested specialists. The final revised manuscript can be found in the uploaded file ***manuscript_with track changes.docx***.

*5) The manuscript lacks an adequate discussion of the limitations of this retrieval technique. This would be essential for readers considering applying this method in different regions or with slightly different instruments.*

**Authors' response:**

We sincerely thank the reviewer for this critical suggestion. We fully agree that a

thorough discussion of the limitations is essential for effective application in different scenarios. In response, we have added a new section titled "***3.4 Limitations and uncertainties***" of our revised manuscript. This section provides a detailed examination of the constraints and potential uncertainties of our proposed retrieval framework. ***Section 3.4 Limitations and uncertainties*** has been presented in the replies of **Major Comments #3**.

*6) The SHAP feature importance analysis raises some questions and methodological concerns. For example, why are specific humidity and relative humidity treated as independent? These are clearly related. There is also lacking a discussion about the definition of and why "geopotential" is so important. Also, it strikes me that the combined SHAP value of extinction, relative humidity, and v-wind being under 50% is relatively low considering they are noted to determine the vertical structure and chemical and physical processes (L410-411).*

**Authors' response:**

We thank the reviewer for these insightful comments and methodological concerns regarding the SHAP analysis. We have revised the manuscript accordingly to address each point.

**Q1**: Why are specific humidity and relative humidity treated as independent?

**A1**: Relative humidity exerts a well-established driving role on $PM_{2.5}$ through its influence on hygroscopic growth, aqueous chemistry, and heterogeneous reactions (Chen et al., 2020). In contrast, specific humidity, which represents the total moisture content of a wet air mass, is more closely linked to the vertical diffusion and wet scavenging of $PM_{2.5}$ (Chatfield et al., 2020). Therefore, including both relative humidity and specific humidity as independent features allows the machine model to leverage their complementary roles in governing $PM_{2.5}$ chemical compositions.

**Q2**: Lacking a discussion about the definition of and why "geopotential" is so important.

**A2**: Geopotential is an integrated feature that reflects the synoptic meteorological conditions when combined with wind fields and is closely related to PM$_{2.5}$ pollution processes, such as accumulation, transboundary transport and dispersion (Jia et al., 2022; Wang et al., 2021). Crucially, synoptic meteorological conditions identified by geopotential patterns largely determine the development of the planetary boundary layer, influencing the vertical distribution of PM$_{2.5}$ (Miao et al., 2022; Xu et al., 2019).

Section 3.2, Line 473-477: "…*Specific humidity (SH) and geopotential (GEOP) also provided important contributions (13.04% and 12.85%, respectively). SH is related to the vertical diffusion and wet scavenging of pollutants (Chatfield et al., 2020) and GEOP identifies the synoptic meteorological patterns that affect both horizontal process (Jia et al., 2022; Wang et al., 2021) and vertical distribution of pollutants within the boundary layer (Miao et al., 2022; Xu et al., 2019).*"

**Reference**

Chatfield, R. B., et al.: Satellite mapping of PM$_{2.5}$ episodes in the wintertime San Joaquin Valley: a "static" model using column water vapor, Atmos. Chem. Phys., 20, 4379-4397, https://doi.org/10.5194/acp-20-4379-2020, 2020.

Chen, Z., et al.: Influence of meteorological conditions on PM$_{2.5}$ concentrations across China: A review of methodology and mechanism, Environ. Int., 139, 105558, https://doi.org/10.1016/j.envint.2020.105558, 2020.

Jia, Z., et al.: The impact of large-scale circulation on daily fine particulate matter (PM$_{2.5}$) over major populated regions of China in winter, Atmos. Chem. Phys., 22, 6471-6487, https://doi.org/10.5194/acp-22-6471-2022, 2022.

Miao, Y., et al.: Influence of Multi-Scale Meteorological Processes on PM$_{2.5}$ Pollution in Wuhan, Central China, Front. Environ. Sci., 10, https://doi.org/10.3389/fenvs.2022.918076, 2022.

Wang, X., et al.: Dominant synoptic patterns associated with the decay process of PM$_{2.5}$ pollution episodes around Beijing, Atmos. Chem. Phys., 21, 2491-2508, https://doi.org/10.5194/acp-21-2491-2021, 2021.

Xu, Y., et al.: Two Inversion Layers and Their Impacts on PM$_{2.5}$ Concentration over the Yangtze River Delta, China, J. Appl. Meteor. Climatol., 58, 2349-2362, https://doi.org/10.1175/JAMC-D-19-0008.1, 2019.

**Q3**: The combined SHAP value of extinction, relative humidity, and v-wind being

under 50% is relatively low considering they are noted to determine the vertical structure and chemical and physical processes.

**A3**: The SHAP values presented in **Section 3.2** were derived from a dataset spanning approximately two years. This long-term perspective reveals that the vertical distribution of PM$_{2.5}$ chemical components is governed by the complex and nonlinear interaction of a multitude of driving features, which contrasts with specific pollution episodes where a single driver may be dominant. From another perspective, our results indicate that the machine learning model effectively captured a complex multi-factorial relationship, rather than relying on an oversimplified representation dependent on a few dominant features. In response to this comment, we have revised the manuscript by replacing "dominant" with the more appropriate term "significant". The revision is as follows.

Section 3.2, Line 459-462: "*Figure 7a1-a5 depicts that the aerosol extinction coefficient at 532 nm (EXT), relative humidity (RH) and v-component wind (VW) are the significant input features for predicting the five PM$_{2.5}$ chemical components with an averaged relative contribution of 14.43 %, 15.84 % and 16.77 %. These features largely affect the vertical structure, chemical and physical processes, respectively…*"

*3 Minor Comments:*

*1) The introduction would benefit from making note of previous work in retrieving PM$_{2.5}$ concentrations from space-based lidar (e.g., Matus et al., 2024; Toth et al., 2022).*

**Authors' response:**

We thank the reviewer for the suggestion. The revised version is as follows.

Introduction, Line 45-47: "…*Continuous remote-sensing lidar detection technologies with high temporal and vertical resolution serve as robust pathways for the constant identification of PM$_{2.5}$ and its components across all altitudes (Matus et al., 2024; Toth et al., 2022; Wang et al., 2022)* …"

**Reference**

Matus, A. V., Nowottnick, E. P., Yorks, J. E., and da Silva, A. M.: Enhancing surface PM$_{2.5}$ air quality estimates in GEOS using CATS lidar data, Earth and Space Sci., 12, e2024EA004078, https://doi.org/10.1029/2024EA004078, 2025.

Toth, T. D., Zhang, J., Vaughan, M. A., Reid, J. S., and Campbell, J. R.: Retrieving particulate matter concentrations over the contiguous United States using CALIOP observations, Atmos. Environ., 274, 118979, https://doi.org/10.1016/j.atmosenv.2022.118979, 2022.

*2) L23: Specify the Chinese megacity (Beijing-Tianjin-Hebei region)*

**Authors' response:**

We thank the reviewer for the suggestion. The revised version is as follows.

Abstract, Line 21-23: "*…Finally, a dataset of vertical mass concentration profiles of these components over six years in a Chinese megacity (Beijing) was generated by the retrieval framework…*"

*3) L41-42: Citing papers for examples of tower, aircraft, balloon, and UAV campaigns is not necessary. These are very common platforms for atmosphere remote sensing.*

**Authors' response:**

We thank the reviewer for the suggestion, and we have removed these citations in the revised manuscript.

Introduction, Line 41-42: "*Field campaigns are widely conducted to obtain vertical profiles of PM$_{2.5}$ chemical components by mounting observation instruments on meteorological towers, aircraft, tethered balloons and unmanned aerial vehicles…*"

*4) L263-264: Specify how these data are averaged.*

**Authors' response:**

We thank the reviewer for the suggestion. The revised version is as follows.

Section 2.1.1, Line 96-98: "*…To generate an hourly resolution lidar dataset, minute-level data were resampled using a simple averaging method. Specifically, the arithmetic*

*mean was calculated from all valid minute-level data points within each non-overlapping one-hour window aligned to the start of each hour (e.g., from 00:00 to 00:59).*"

*5) Figure 3: Specify the observations used in the figures. All altitudes from the tower and airplane?*

**Authors' response:**

We thank the reviewer for the suggestion. To clarify, ***Fig. 3*** exclusively uses ground-level observations to independently validate the ground-level predictions from our machine learning model, as the model itself was trained and tested exclusively on surface data. The tower-based and aircraft-based measurements were reserved for the independent validation of the retrieved vertical profiles, which is presented in ***Fig 6***. In response to the suggestion, the revised version is as follows.

Section 2.2.4, Line 322-327: "*An hourly multivariate dataset with extensive temporal coverage was employed to train and evaluate the deep learning module. To maintain temporal independence, the training (and validation) set was constructed from a 1-year (2021) time-series dataset obtained from a Beijing site (Fig. S1), while the testing set contains an independent 6-month (Jan 1-Mar 31 and Jun 1 to Aug 31, 2022) time-series dataset obtained from the same site. A 10-fold time-series cross-validation (CV) scheme was designed for the training (and validation) set to preserve its temporal order and prevent future information leakage, which is detailed in Text S3 and Fig. S2 of the supplement. The iteration number of Bayesian optimization is set to 20.*"

Section 3.1.1, Line 345-347: "*The 10-fold CV sets and a testing set with temporal independence are utilized to evaluate the predictive performance of the deep learning module, which is quantified by the discrepancies between simulations and observations at ground level for $NH_4^+$, $SO_4^{2-}$, $NO_3^-$, OM and BC…*"

*6) Figure 4: There are many cases where the retrieval is not particularly close to the observations. A broader discussion about these cases would be valuable.*

**Authors' response:**

We thank the reviewer for the suggestion and have added a broader discussion in the revised manuscript as follows. Besides, the discussion in retrieval uncertainties has been presented in the replies of **Major Comments #3**.

Section 3.1.2, Line 385-392: "*…These results indicate that the retrieval framework roughly interprets the changes in concentrations of various chemical components across different periods, exhibiting fundamental temporal generalization capabilities. However, the retrieved concentrations show some overestimation cases during autumn in 2018 and spring in 2024, potentially associated with the uncertainties induced by the training data. The training data may lack a sufficiently diverse spectrum of meteorological conditions and pollution patterns, which limits the temporal generalizability of the retrieval framework across all complex and dynamic atmospheric scenarios. Future efforts should enhance retrieval accuracy by augmenting the training data with observations spanning a wider range of temporal conditions.*"

*7) Figure 5: Subplot "a" needs a better explanation. What is conveyed differently in the histograms versus the dots? Subplots in the "b" and "c" rows would benefit from a better map. Readers from outside China may be lost without other context (coloring the ocean/seas, highlighting major cities, etc.)*

**Authors' response:**

Figure 5a highlights the data distribution properties derived from the retrieved and observed surface mass concentration of $NH_4^+$, $NO_3^-$, $SO_4^{2-}$, OM and BC at 39 non-training BTH lidar sites over a period of February 8-15th, 2021. Figure 5a combines dotplots, boxplots and Kernel density. Kernel density highlights the overall shape of the data distribution. Dotplots, while similar in Kernel density, also convey information about the exact number of datapoints across the distribution.

In response to the Reviewer's comments, we have removed the dotplots. Because the high density of data points resulted in a cluttered presentation, and the information

they convey about the distribution is effectively captured by the kernel density, which provides a smoother and more interpretable representation. Then we added geographic basemaps into Figure 5b, c for better presentation.

[Figure]

*Figure 5: Data distribution properties of retrieved and observed surface mass concentration (µg m⁻³) of NH₄⁺, NO₃⁻, SO₄²⁻, OM and BC at 23 non-training NCP lidar sites over a period of February 8-15ᵗʰ, 2021, presented by a combination of boxplots and kernel density (a). Spatial distribution of Pearson correlation coefficient (CORR) between retrieved and observed surface mass concentration of NH₄⁺ (b1), NO₃⁻ (b2), SO₄²⁻ (b3), OM (b4) and BC (b5). (c1-c5) Same as (b1-b5) but for root mean square error (RMSE, µg m⁻³). The geographic basemap is hosted by Esri (https://www.esri.com/en-us/home).*

*8) L418: Define "upper atmosphere" in this context*

**Authors' response:**

We thank the reviewer for the reminder. The term "upper atmosphere" is inaccurate, and we have replaced with "upper planetary boundary layer" in the revised manuscript.

Section 3.2, Line 469-470: "...*The aerosol content in the upper planetary boundary layer is relatively low...*"

Section 3.2, Line 471-472: "...*Conversely, pollution transport in the upper planetary boundary layer is less affected by interference from complex underlying surfaces than near-surface transport...*"

---

## Author Comment (AC2)

**Authors' responses to Referees' comments**

**Journal:** Atmospheric Measurement Techniques

**Manuscript Number:** egusphere-2025-4237

**Title:** A Physics-Constrained Deep-Learning Framework based on Long-Term Remote-Sensing Data for Retrieving Vertical Distribution of PM$_{2.5}$ Chemical Components

**Authors:** Hongyi Li, Ting Yang, et al.

Note:

Comment (12-point black italicized font).

Reply (indented, 12-point blue normal font).

"Revised text as it appears in the text (in quotes, 12-point blue italicized font)".
* * *
**Anonymous Referee #2**

*1 General comments:*

*This manuscript presents a novel lidar-based retrieval framework that integrates deep learning with physics-constrained optimization to estimate vertical mass concentration profiles of five PM$_{2.5}$ chemical components (SO$_4^{2-}$, NO$_3^-$, NH$_4^+$, OM, BC). The topic is scientifically important: retrieving aerosol composition profiles from lidar would significantly advance air quality monitoring, chemical transport modeling, and source apportionment. The combination of physics constraints and deep learning is innovative and promising.*

*However, while the conceptual idea is strong, the manuscript lacks clarity in describing the model framework and provides insufficient evidence that the approach accurately captures the physics of vertical aerosol composition or generalizes across seasons, sites, and aerosol regimes. Significant issues in methodology, validation, and presentation hinder the scientific interpretation of the results. I therefore recommend major revision.*

**Authors' response:**

We extend our sincere gratitude to the reviewer for the insightful and constructive evaluation of our manuscript, which has been instrumental in improving our work. We are particularly encouraged by the reviewer's recognition of the scientific importance of our topic and the innovation. We fully agree with the reviewer's assessment that,

while the conceptual idea is strong, the manuscript in its current form requires major revisions to achieve the necessary clarity and evidentiary rigor. We have implemented a point-by-point revision of the manuscript in direct response to the reviewer's concerns and comments.

*2 Major Comments:*

*1) Overall framing, workflow clarity, and Figure 1*

*Q1*: *Figure 1 is difficult to interpret: inputs/outputs are not clearly labeled, colored boxes lack explanation, and several acronyms are undefined. The figure should be redesigned as a clear block-flow diagram that lists:*

*a) all inputs (with units, vertical resolution, and dimensionality),*

*b) each module's output,*

*c) loss functions used,*

*d) data flow direction and optimization loops.*

*Q2*: *Reorder sections so the Data section precedes the Model description. Readers must understand what data the model consumes before interpreting architectural choices.*

**Authors' response:**

**A1**: We thank the reviewer for this critical suggestion. In response, Figure 1 has been completely redesigned to enhance its interpretability. The revised Fig. 1 contains all inputs with specific information, two modules' output, loss functions used, as well as data flow direction and optimization loops. The abbreviations used in Fig.1 have been defined in the figure caption. Notably, since the intermediate output of the deep learning module serves as the input data for the optimization module, the color of deep learning output is same as that of input data.

[Figure]

**Figure 1**: *Remote-sensing retrieval framework for vertical distribution of five PM$_{2.5}$ chemical components (NH$_4^+$, SO$_4^{2-}$, NO$_3^-$, OM and BC). (U: U-component wind; V: V-component wind; T: Temperature; RH: Relative Humidity; q: Specific Humidity; w: Vertical Velocity; Z: Geopotential; $\sigma_{bsc,532}$: Aerosol Extinction Coefficient at 532 nm; CNN: Convolutional Neural Network; ReLU: Rectified Linear Unit; FC: Fully Connected; BiLSTM: Bidirectional Long Short-Term Memory; IMPROVE: Interagency Monitoring of Projected Visual Environment; NSGA-II: Non-dominated Sorting Genetic Algorithm II).*

**A2**: Following the reviewer's suggestions, we have reordered the "***Section 2 Data and methodology***". In the revised manuscript, ***Section 2.1*** details the data information and ***Section 2.2*** details the methodologies used in this work.

*2) Ambiguity in algorithm description*

**Q1**: *Section 2.1.1 is confusing and lacks foundational background, making the workflow difficult to follow without jumping back and forth.*

**Q2**: *It is unclear what the deep-learning model predicts per vertical level. Please explicitly specify:*

*a) whether the model outputs component concentrations, component fractions, categorical flags, or something else,*

*b) the exact dimensionality (e.g., levels $\times$ 5 components).*

**Q3**: *Clearly define the target variables and how they are constructed.*

**Q4**: *Provide detailed descriptions of the multi-objective optimization, including:*

*a) inputs and outputs,*

*b) spatial/temporal/vertical resolution,*

*c) how physics constraints are incorporated mathematically.*

**Q5**: *The purpose of using mentioned components/models.*

**Q6**: *The rationale for using a two-step prediction process (component "flags" followed by concentrations) rather than a single multi-output network is not explained. The manuscript would benefit from an experimental justification or comparison.*

**Authors' response:**

We thank the reviewer for highlighting the lack of clarity in Section 2.1.1. We have completely revised "***Section 2.2.1 Retrieval Framework***" in response to the reviewer's concerns regarding above six aspects. Notably, the original Section 2.1.1 has been moved to ***Section 2.2.1*** in response to **Major Comments #1**.

**A1**: We have thoroughly revised the description of ***Section 2.2.1*** and added the description of foundational backgrounds. The revised ***Section 2.2.1*** is presented below.

Section 2.2.1: *"This paper proposed a novel retrieval framework for retrieving the vertical concentration profiles of five $PM_{2.5}$ chemical components ($NH_4^+$, $SO_4^{2-}$, $NO_3^-$, OM and BC) from the lidar aerosol extinction coefficient at 532 nm ($\sigma_{bsc,532}$). As shown in Fig. 1, the retrieval framework mainly consists of a deep learning module and a physics-constrained optimization module. The input datasets of the deep learning*

*module include the surface observation data, meteorological data and ground-based lidar data (Fig. 1a). Specifically, the aerosol extinction coefficient at 532 nm ($\sigma_{bsc,532}$) and multiple meteorological parameters (u-component wind, v-component wind, temperature, relative humidity, specific humidity, vertical velocity and geopotential) serve as input features, while the concentrations of the five $PM_{2.5}$ chemical components ($NH_4^+$, $SO_4^{2-}$, $NO_3^-$, OM and BC) serve as target features. The deep learning module (Fig. 1b), mainly consisting of the Convolutional Neural Network (CNN), Bidirectional Long Short-Term Memory (BiLSTM), attention mechanism and Bayesian optimization, is utilized to establish the nonlinear relationship between input and target features. The input datasets of the physics-constrained optimization module include the ground-based lidar data, aerosol auxiliary data and deep learning intermediate output (Fig. 1a, c), which provide fundamental input for establishing a multi-object function based on the Interagency Monitoring of Projected Visual Environment (IMPROVE) equation. The physics-constrained optimization module incorporates the multi-object loss function with the Non-dominated Sorting Genetic Algorithm II (NSGA-II) to implement external physical constraints (Fig. 1c), thus enhancing the extrapolation capability of the deep learning module and generating high-quality vertical concentration profiles of the five $PM_{2.5}$ chemical components. Detailed descriptions of the deep learning algorithms, hyperparameter tuning, and physics-constrained optimization used in this work will be presented in subsequent sections. The brief workflow of the retrieval framework is summarized as follows.*

*Step 1. The multi-source input datasets undergo matching across spatiotemporal and vertical dimensions. All input and output data are uniformly time-resolved to hourly intervals, while vertical data are uniformly vertically resolved into 10 layers ranging from 50 m to 3 km.*

*Step 2. The input data of the deep learning module are normalized by Z-score normalization to stabilize the training process, accelerate training convergence, and enhance model robustness (Al-Faiz et al., 2018; Cabello-Solorzano et al., 2023).*

*Step 3. Training deep learning module by using the normalized surface-level input data.*

*Step 4. Generating the normalized concentrations of the five PM$_{2.5}$ chemical components at each vertical layer by feeding the normalized height-level input data into the deep learning module.*

*Step 5. Denormalizing the deep-learning output by using the inverse Z-score transformation, with the mean and standard deviation statistics derived from the original training set, thereby recovering the physical mass concentration unit (μg m$^{-3}$).*

*Step 6. Optimizing the denormalized deep learning output through implementing an external physics constraint to obtain the high-quality vertical concentration profiles of the five PM$_{2.5}$ chemical components. Repeat steps 4-6 until the retrieval task is complete.*"

**A2**: In response to the reviewer's suggestion, we have specified the final output (the vertical concentration profiles of the five PM$_{2.5}$ chemical components) and its exact dimensionality ([Height, Timestep, Variable, Site]) in the revised ***Fig. 1*** and ***Section 2.2.1***, which were presented in the replies of **Major Comments #1** and **Major Comments #2 A1**, respectively.

**A3**: In response to the reviewer's suggestion, we have clearly defined the target variables and how they are constructed in the revised ***Fig. 1*** and ***Section 2.2.1***, which were presented in the replies of **Major Comments #1** and **Major Comments #2 A1**, respectively.

**A4**: In response to the reviewer's suggestion, we have added detailed descriptions of the input and output of the multi-objective optimization module with their temporal and vertical resolutions in the revised ***Fig. 1*** and ***Section 2.2.1***, which were presented in the replies of **Major Comments #1** and **Major Comments #2 A1**, respectively. Detailed descriptions of how physics constraints are incorporated mathematically have been added in the revised ***Section 2.2.3***. The revised ***Section 2.2.3*** is presented below.

: *"The normalized vertical profiles of PM$_{2.5}$ chemical components generated by the deep learning module are denormalized by the statistical characteristics of the initial input data of the surface-level observations. To reduce the retrieval error induced by the inherent extrapolation limitations of deep learning modules, a physics-constrained optimization scheme is incorporated into the retrieval framework based on a revised Interagency Monitoring of Projected Visual Environment (IMPROVE) Equation (Pitchford et al., 2007) and Non-dominated Sorting Genetic Algorithm II (NSGA-II) (Verma et al., 2021).*

*The revised IMPROVE Equation interprets the particle extinction coefficient ($\sigma$) through the concentrations (M) and the optical and microphysical characteristics of PM$_{2.5}$ chemical components (Eq. (7)).*

$$\sigma(M) = \theta_s^{SNA} f(RH)(M(SO_4^{2-}) + M(NO_3^-) + M(NH_4^+)) + \theta_s^{OC} M(OC) +$$
$$\theta_s^{FS} M(Fine\ Soil) + \theta_s^{CM} M(Coarse\ Mass) + \theta_s^{FSS} f_{FSS}(RH) M(Fine\ Sea\ Salt) +$$
$$\theta_a^{BC} M(BC) + Rayleigh\ Scattering, \qquad (7)$$

*Where $\sigma(M)$ is the estimated particle extinction coefficient (km$^{-1}$), $\theta_s$ is the scattering efficiency (m$^2$ mg$^{-1}$), $\theta_a$ is the mass absorption efficiency (m$^2$ mg$^{-1}$), respectively. $f(RH)$ and $f_{FSS}(RH)$ account for the increase in light scattering induced by hygroscopic growth of sulfate, nitrate and ammonium (SNA), as well as fine sea salt (FSS). $\theta_s^{FS}$, $\theta_s^{CM}$, $\theta_s^{FSS} f_{FSS}$ and $\theta_a^{BC}$ are set to 0.001 m$^2$ mg$^{-1}$, 0.0006 m$^2$ mg$^{-1}$, 0.0017 m$^2$ mg$^{-1}$ and 0.01 m$^2$ mg$^{-1}$, respectively. M are the mass concentrations (µg m$^{-3}$) of the PM$_{2.5}$ chemical components. Rayleigh Scattering is set to 0.01 km$^{-1}$. $\theta_s^{SNA}$ and $\theta_s^{OC}$ are determined by Eq. (8)-(9).*

$$\theta_s^{SNA} = 0.003 \times (0.7 + 0.002 \times (M(SO_4^{2-}) + M(NO_3^-) + M(NH_4^+) + M(OC)])),$$
$$(8)$$

$$\theta_s^{OC} = 0.00363 \times (0.7 + 0.002 \times (M(SO_4^{2-}) + M(NO_3^-) + M(NH_4^+) + M(OC))) \quad ,$$
$$(9)$$

*To implement the physics-constrained optimization, we first introduce a scale factor*

*($\gamma_{i,h}$) for each chemical component at each vertical layer, which is used to correct the initial mass concentrations (Eq. (10)). Then we determine the optimal scale factors through minimizing a multi-objective function (Eq. (11)). The Pearson correlation coefficient (CORR) and root mean square error (RMSE) quantified by the lidar-observed and the IMPROVE-simulated extinction coefficient serve as two objective values in the multi-objective function. The NSGA-II algorithm is utilized to determine the optimal scale factors by solving the multi-objective function that simultaneously enhances the correlation and reduces the discrepancy between the IMPROVE-estimated and lidar-observed extinction coefficients.*

$$M_{regulated}^{i,h} = \gamma_{i,h} \times M_{original}^{i,h}, i = SO_4^{2-}, NO_3^-, NH_4^+, OM, and\ BC,$$

*(10)*

$$\gamma_{i,h} = min(f_{RMSE}(\gamma), f_{CORR}(\gamma)), \tag{11}$$

*Where $M_{regulated}^{i,h}$ (µg m$^{-3}$) is the regulated mass concentration of the $i^{th}$ chemical component at an altitude of h (m), $\gamma_{i,h}$ is the scale factor for the $i^{th}$ chemical component at an altitude of h (m), and $M_{original}^{i,h}$ (µg m$^{-3}$) is the original mass concentration of the $i^{th}$ chemical component at an altitude of h (m). $f_{RMSE}(\gamma)$ is the RMSE-based objective function (Eq. (12)) and $f_{CORR}(\gamma)$ is the CORR-based objective function (Eq. (13)).*

$$f_{RMSE}(\gamma) = \sqrt{\frac{\sum_{k=1}^{K}(\sigma_k^{obs} - \sigma_k(\gamma \times M))^2}{K}}, \tag{12}$$

$$f_{CORR}(\gamma) = -\frac{\sum_{k=1}^{K}(\frac{\sigma_k(\gamma \times M) - \overline{\sigma(\gamma \times M)}}{std(\sigma(\gamma \times M))})(\frac{\sigma_k^{obs} - \overline{\sigma^{obs}}}{std(\sigma^{obs})})}{K-1}, \tag{13}$$

*Where K is the total number of samples, $\sigma_k^{obs}$ is the $k^{th}$ observed extinction coefficient, $\sigma_k(\gamma \times M)$ is the $k^{th}$ simulated extinction coefficient, $\overline{\sigma(\gamma \times M)}$ is the average of simulated extinction coefficient, $\overline{\sigma^{obs}}$ is the average of observed extinction coefficient, $std(\sigma(\gamma \times M)$ is the standard deviation of simulated extinction coefficient, and $std(\sigma^{obs})$ is the standard deviation of observed extinction coefficient."*

**A5**: In response to the reviewer's suggestion, we have clarified the purpose of the

deep learning module and the physics-constrained optimization module in the revised *Section 2.2.1*, which were presented in the replies of **Major Comments #2 A1**.

**A6**: Performing a two-step prediction process (normalization → model inference → denormalization) is a standard and necessary practice in our deep learning module. Disparities in the scales and units of different features would cause those with larger numerical ranges to dominate gradient updates, hindering the learning of complex interactions. Data normalization can stabilize the training process, accelerate training convergence, and enhance model robustness (Al-Faiz et al., 2018; Cabello-Solorzano et al., 2023). Accordingly, the data of input and target features used for the deep learning module are normalized in this study. As a result, the deep learning module initially outputs normalized concentrations (like "flags"), which are subsequently denormalized and optimized to yield high-accuracy mass concentrations.

In response to the reviewer's suggestions, we have emphasized the role of data normalization in the revised *Section 2.2.1*, which were presented in the replies of **Major Comments #2 A1**.

**Reference**

Al-Faiz, M. Z., Ibrahim, A. A., and Hadi, S. M.: The effect of Z-Score standardization (normalization) on binary input due the speed of learning in back-propagation neural network, Iraqi j. inf. commun. technol., 1, 42-48, https://doi.org/10.31987/ijict.1.3.41, 2018.

Cabello-Solorzano, K., Ortigosa de Araujo, I., Peña, M., Correia, L., and J. Tallón-Ballesteros, A.: The Impact of Data Normalization on the Accuracy of Machine Learning Algorithms: A Comparative Analysis, 18th International Conference on Soft Computing Models in Industrial and Environmental Applications (SOCO 2023), Cham, 344-353, https://doi.org/10.1007/978-3-031-42536-3_33, 2023.

**3) Temporal and spatial data splits**

*Q1: The current random 80/20 split is not appropriate for meteorological/aerosol time series due to temporal autocorrelation, which risks information leakage.*

*Q2: Consider implementing:*

*a) temporal holdouts (e.g., full seasons),*

*b) spatially independent test sites,*

*c) blocked k-fold cross-validation preserving temporal/spatial independence.*

*Q3*: *The manuscript evaluates an independent dataset only in the Results section, but this dataset should be partially used for the validation/testing framework.*

*Q4*: *The reported error statistics for the independent dataset are not clearly presented and differ considerably from training results. For a well-generalized model, validation and independent-test errors should be similar; their discrepancies raise concerns about generalization and physical consistency.*

*Q5*: *Surface-only scatterplots from the training year are insufficient to establish model validity, especially given that the model's primary output is a vertical distribution.*

**Authors' response:**

We sincerely thank the reviewer for these critical and constructive comments on the data split strategy.

**A1**: We fully agree that the random split is not appropriate for the time-series training data, especially for meteorological and aerosol data, since future information leakage could lead to artificially optimistic testing results. Following the reviewer's suggestions, we have repartitioned the dataset into training (& validation) and independent testing sets. Detailed description of data repartition can be found in the replies of **Major Comments #3 A2**.

**A2**: In response to the reviewer's suggestions, the repartitioned testing set contains an independent 6-month (Jan 1-Mar 31 and Jun 1 to Aug 31, 2022) time-series dataset obtained from a Beijing site for ensuring data temporal independence. The repartitioned training (& validation) set contains an independent 1-year (2021) time series dataset obtained from a Beijing site.

A spatially independent 8-day (Feb 8-15, 2021) time-series dataset obtained from 23 sites in the North China Plain (NCP) is utilized to evaluate the spatial extrapolation capability of our retrieval framework. The spatial distribution of training (& validation) sites, temporally independent testing sites, and spatially independent testing sites is presented in *Fig. S1* of the supplement.

[Figure]

***Figure S1****: Spatial distribution of Beijing and spatially independent testing sites. Beijing site provides 18-month datasets for the training, validation and temporally independent testing of the deep-learning module. Other 23 sites in North China Plain (NCP) provide 8-day datasets for the spatially independent testing of the final retrieval. The geographic basemap is hosted by Esri ([https://www.esri.com/en-us/home](https://www.esri.com/en-us/home)).*

A blocked k-fold cross-validation may leak future information into the training set while allocating historical information to the validation set in practice. In response to the reviewer's suggestions, **we replaced the original random split scheme with a time-series cross-validation scheme to preserve temporal order and prevent future information leakage**. As presented in ***Fig. S2*** of the supplement, we repeatedly utilize a forward sliding window to create K (set to 10) validation folds. The training set starts with a subset of the first 80% of the chronological data and is incrementally expanded at each subsequent fold by incorporating an additional block with a length of the forward sliding window, ultimately encompassing the full 80% in the final fold. The validation set immediately follows the training set, comprising 20% of the

chronological data.

The length of the forward sliding window is equal to the length of the training set at first fold in practice (Eq. (R1)).

$$l = \frac{r \times N}{K},$$ (R1)

where $l$ is the length of the forward sliding window, $r$ is the proportion of data used for training, $N$ is the total sample size for model construction, and $K$ is the total number of cross-validation folds.

[Figure]

Figure S2: Diagram of the time-series data partitioning for 10-fold cross-validation.

**A3**: In response to the reviewer's suggestions, we have used the temporally independent data for the validation and testing phases. The detailed description can be found in the replies of **Major Comments #3 A2**. The revised version is as follows.

Section 2.2.4, Line 345-354: "*An hourly multivariate dataset with extensive temporal coverage was employed to train and evaluate the deep learning module. To maintain temporal independence, the training (and validation) set was constructed from a 1-year*

*(2021) time-series dataset obtained from a Beijing site (Fig. S1), while the testing set contains an independent 6-month (Jan 1-Mar 31 and Jun 1 to Aug 31, 2022) time-series dataset obtained from the same site. A 10-fold time-series cross-validation (CV) scheme was designed for the training (and validation) set to preserve its temporal order and prevent future information leakage, which is detailed in Text S3 and Fig. S2 of the supplement. The iteration number of Bayesian optimization is set to 20."*

**A4**: We thank the reviewer for the reminder regarding the clarification in comparison between validation and independent-test error statistics. The revised data repartition scheme is detailed in the replies of **Major Comments #3 A2**. Following the reviewer's suggestion, we have verified our deep learning model using the new data repartition scheme. Finally, we have revised ***Section 3.1.1***, updated ***Fig. 3***, and added ***Fig. S3*** to comprehensively present and discuss the error statistics in the 10-fold cross-validation phase and temporally independent testing phase.

The results show that the error distributions are concentrated around 0, with mean errors between $-1.78 \pm 8.15$ µg m$^{-3}$ and $-0.13 \pm 0.94$ µg m$^{-3}$ during the 10-fold CV phase (***Fig. S3a1-a5***) and between $-1.36 \pm 7.40$ µg m$^{-3}$ and $-0.07 \pm 1.00$ µg m$^{-3}$ during the temporally independent testing phase (***Fig. S3b1-b5***), demonstrating strong consistency between observations and simulations. **Notably, the error distributions for the validation and independent testing sets are closely aligned, indicating that the deep learning module is robust and generalizes well to unseen data.**

Section 3.1.1, Line 345-363: *"The 10-fold CV sets and a testing set with temporal independence are utilized to evaluate the predictive performance of the deep learning module, which is quantified by the discrepancies between simulations and observations at ground level for $NH_4^+$, $SO_4^{2-}$, $NO_3^-$, OM and BC. Overall, the scatter distribution and fitted regression line closely align with the 1:1 line in both the 10-fold CV (Fig. 3a1-a5) and temporally independent testing phases (Fig. 3b1-b5). The error distributions are concentrated around 0, with mean errors between $-1.78 \pm 8.15$ µg m$^{-3}$ and $-0.13 \pm 0.94$ µg m$^{-3}$ during the 10-fold CV phase (Fig. S3a1-a5) and between -*

*1.36 ± 7.40 µg m⁻³ and -0.07 ± 1.00 µg m⁻³ during the temporally independent testing phase (Fig. S3b1-b5), demonstrating strong consistency between observations and simulations. Notably, the error distributions for the validation and independent testing sets are closely aligned, indicating that the deep learning module is robust and generalizes well to unseen data. Specifically for the 10-fold CV process (Fig. 3a1-a5), the CORR values for the five PM₂.₅ chemical components range from 0.76 to 0.86, indicating that the deep learning module accurately interprets the relationship between multivariate input features and the five PM₂.₅ chemical components. The RMSE values range from 0.95 to 8.35 µg m⁻³, indicating a low discrepancy between simulations and observations. Compared to the 10-fold CV process, the temporally independent testing yields slightly lower CORR values (0.69-0.79) and higher RMSE values (1.00-8.87 µg m⁻³), showing a slight underestimation for the five PM₂.₅ chemical components (Fig. 3b1-b5). It is expected that the statistical results from the temporally independent testing are less robust than those from the 10-fold CV, since the temporally independent testing set aggregates a broader spectrum of temporal patterns compared to the validation set at each fold. Our statistical results from the 10-fold CV exhibit similarities or even improvements compared to those reported in other studies that predicting PM₂.₅ chemical component concentrations based on machine learning models (Lv et al., 2021; Lin et al., 2022; Araki et al., 2022; Liu et al., 2023), indicating that the deep learning module demonstrates strong prediction capabilities."*

[Figure]

*Figure 3: Scatterplots of the simulations (µg m⁻³) versus the observations (µg m⁻³) with probability density (%) for NH₄⁺, NO₃⁻, SO₄²⁻, OM and BC during the 10-fold cross-*

*validation process (a1-a5) and temporally independent testing process (b1-b5). The dotted grey lines represent the 2:1, 1:1, and 1:2 lines, and the solid red line represents the fitted regression line. CORR represents the correlation coefficient, and RMSE represents root mean square error.*

[Figure]

***Figure S3**: Probability distributions of error (observations minus simulations, µg m⁻³) for NH₄⁺, NO₃⁻, SO₄²⁻, OM and BC during the 10-fold cross-validation phase (a1-a5) and during the temporally independent testing phase (b1-b5). ME: Mean Error; SD: Standard Deviation.*

**A5**: We fully agree that including scatter plots at multiple vertical levels would provide a more comprehensive validation of the model's vertical retrieval capability. However, conducting long-term vertical measurements (especially covering altitudes of 0-3 km) of $PM_{2.5}$ chemical compositions is exceptionally challenging. Vertical measurements typically rely on costly airborne measurements (e.g., aircraft or balloon-borne instruments), which are neither continuous at a fixed location nor provide broad spatial coverage over extended periods. Given the scarcity of vertical samples, we instead obtained a long-term and spatially extensive dataset of ground-level $PM_{2.5}$ chemical observations. The sufficient ground-level dataset was used to train the deep neural network in establishing the nonlinear mapping between lidar extinction coefficients and chemical component concentrations. Consequently, the scatter-plot validation focuses primarily on the ground level. We have added a dedicated ***Section 3.4 Limitations and uncertainties*** in the revised manuscript to discuss the current

limitations regarding vertical measurement validation. The content of ***Section 3.4 Limitations and uncertainties*** is presented in the replies of **Major Comments #6**.

**4) Weak vertical-profile validation**

*The manuscript focuses on retrieving vertical composition profiles but presents minimal validation of these profiles. I strongly recommend including:*

***Q1****: Direct comparisons with aircraft or in situ vertical measurements, using metrics such as bias, RMSE, MAE, percent error, and correlation at each altitude bin.*

***Q2****: Case studies across representative aerosol regimes (smoke, dust, pollution, background).*

***Q3****: Aggregated statistics by:*

*a) altitude,*

*b) site,*

*c) aerosol type,*

*d) season.*

*If vertical observational data are limited, the manuscript should explicitly quantify these limitations while still presenting as much vertical validation as possible.*

**Authors' response:**

We fully acknowledge the reviewer's point that robust validation of the retrieved vertical profiles is essential, and we agree that expanding the vertical validation with detailed metrics, case studies, and aggregated statistics would be ideal. However, as the reviewer anticipated, providing extensive vertical validation is inherently constrained by vertical data availability.

**A1**: In response to the reviewer's suggestion, we have added ***Table S3*** to present statistical metrics quantified by vertical retrievals and tower-based observations during a period from December 30, 2018 to January 2, 2019. Besides, we have updated ***Fig. 6*** and revised ***Section 3.1.3*** using the revised retrieval framework.

Section 3.1.3: "*In addition to the spatiotemporal verification of surface-level mass concentrations, tower-based and aircraft-based observational experiments were conducted to validate the retrieved vertical profiles of five $PM_{2.5}$ chemical components during non-training periods. From the surface to ~200 m altitude, the retrieved and*

*observed vertical profiles exhibit similar vertical patterns during a period from December 30, 2018 to January 2, 2019 in Beijing, with higher concentrations occurring at altitudes of 50-80 m for $NH_4^+$, $NO_3^-$, $SO_4^{2-}$ and OM (Fig. 6a1, a2). Specifically, as presented in Table S3, the CORR values are no less than 0.66 for all four $PM_{2.5}$ chemical components. However, the RMSE value for OM (23.04 μg $m^{-3}$) is notably higher than that for the other components (4.08-10.48 μg $m^{-3}$), indicating limitations in the retrieval framework when representing the vertical profile of OM during winter pollution episodes. This discrepancy may be associated with retrieval uncertainties arising from input data quality and imposed physical constraints. Additionally, the retrieved and observed proportions of $NH_4^+$, $NO_3^-$, $SO_4^{2-}$, OM and BC demonstrate significant consistency (Fig. 6b1, b2). Among these chemical components, $NO_3^-$ and OM contribute the largest proportions, followed by $NH_4^+$ and $SO_4^{2-}$, while BC contributes the smallest fraction. This proportional characteristic is evident in both the retrieved and observed proportions at altitudes of 600 m and 1200 m (Fig. 6c1, c2). Due to the lack of $NH_4^+$ measurements at 1500 m and the absence of both $NH_4^+$ and $SO_4^{2-}$ measurements at 2100 m, the proportions at these altitudes are statistically inferred from the remaining chemical components. The results indicate overall consistency between retrieved and observed proportions at altitudes of 1500 m and 2100 m, although the proportion of $NO_3^-$ is slightly overestimated at 2100 m and underestimated at 1500 m. Overall, the tower-based and aircraft-based verifications indicate that the retrieval framework achieves high accuracy in retrieving the vertical profiles of the five $PM_{2.5}$ chemical components during non-training period, demonstrating its robust generalization capability and reliability when applied to independent datasets.*"

[Figure]

***Figure 6****: Vertical profiles (µg m⁻³) of NH₄⁺, NO₃⁻, SO₄²⁻, and OM from retrieval (a1) and tower-based observation (a2) during a period from December 30, 2018 to January 2, 2019 in Beijing. The line represents the daily average of the hourly vertical profiles, and the shaded area represents the standard deviation. Averaged proportions of NH₄⁺, NO₃⁻, SO₄²⁻, OM, and BC from retrieval (b1) and tower-based observation (b2) for 10 days (December 27 and 30, 2023; January 2, 5, 9, 12, 15, 18, 24, and 27, 2024). (c1 and c2) Same as (b1 and b2) but for aircraft-based verification for 3 days (September 26, October 10, December 11, 2024).*

*Table S3. Statistical metrics quantified by the vertical retrievals and the tower-based observations during a period from December 30, 2018 to January 2, 2019 for NH₄⁺, NO₃⁻, SO₄²⁻, and OM. RMSE: Root Mean Square Error; MAE: Mean Absolute Error; CORR: Pearson correlation coefficient.*

| | RMSE (µg m⁻³) | MAE (µg m⁻³) | CORR |
|---|---|---|---|

| | | | |
|---|---|---|---|
| NH$_4^+$ | 4.81 | 3.14 | 0.67 |
| NO$_3^-$ | 1048 | 6.19 | 0.67 |
| SO$_4^{2-}$ | 4.08 | 2.59 | 0.66 |
| OM | 23.04 | 15.37 | 0.67 |

**A2 & A3**: The available vertical observational dataset was limited to a 3-day aircraft campaign at a site near Beijing and a 10-day tower measurement at a site in Beijing, both conducted primarily during winter. The scarcity of vertical observational data precludes a comprehensive statistical evaluation of the vertical retrievals with respect to varying sites, aerosol types, and seasons. In response to the reviewer's suggestion, we have added the limitations into *Section 3.4 Limitations and uncertainties* of the revised manuscript. The content of *Section 3.4 Limitations and uncertainties* is presented in the replies of **Major Comments #6**.

*5) Heterogeneous site performance*

*Q1*: *Figure 5b shows substantial site-to-site variability: some sites have nearly zero correlation, while the best site reaches ~0.6.*

*Q2*: *Please investigate and report potential causes, such as:*

*a) aerosol-type mismatch,*

*b) representativeness of training data,*

*c) site-specific meteorology or emissions,*

*d) instrument characteristics.*

*Q3*: *Consider:*

*a) a map showing training vs. test sites,*

*b) per-site metrics (MAE, RMSE, bias, percent error, N),*

*c) problematic site scatterplots or boxplots to illustrate error spread.*

**Authors' response:**

We thank the reviewer for the constructive suggestions.

**A1 & A2**: The observed variability in retrieval performance across different sites, with correlation coefficients ranging from near zero to ~0.6, is primarily attributed to the spatial representativeness of training data. The deep-learning module was trained exclusively on a long-term dataset from a single site in Beijing. A single-site dataset is

insufficient to interpret the varying emission intensity, local meteorological and geographical conditions across the broader Northern China Plain, which limits the spatial extrapolation capability of the deep-learning module. We have explicitly discussed this remaining limitation in **Section 3.1.2** and **Section 3.4**. The content of **Section 3.4 Limitations and uncertainties** is presented in the replies of **Major Comments #6**.

Section 3.1.2, Line 399-414: "*The retrieval framework was also applied to retrieve the vertical profiles of the five $PM_{2.5}$ chemical components at 23 non-training NCP lidar sites over a short-term period of February 8-15th, 2021, aiming to validate its spatial generalization capabilities. Compared with the observed surface concentrations at 23 non-training sites, the retrieved surface concentrations exhibit a more clustered data distribution and exhibit a tendency toward underestimation across all components (Fig. 5a). The site-averaged CORR values for the five chemical components range from 0.21 to 0.46, with RMSE values spanning 2.7 µg m$^{-3}$ to 20.37 µg m$^{-3}$ (Fig. S4). From a spatial perspective (Fig. 5b1-b5), non-training NCP sites located closer to the Beijing lidar site exhibit higher CORR values, with the highest reaching 0.71 ($NH_4^+$), 0.56 ($NO_3^-$), 0.81 ($SO_4^{2-}$), 0.48 (OM) and 0.41 (BC). Conversely, the RMSE values are not affected by the distance from the Beijing lidar site (Fig. 5c1-c5), with the lowest reaching 2.91 µg m$^{-3}$ ($NH_4^+$), 6.15 µg m$^{-3}$ ($NO_3^-$), 3.05 µg m$^{-3}$ ($SO_4^{2-}$), 6.59 µg m$^{-3}$ (OM) and 0.78 µg m$^{-3}$ (BC). However, several sites exhibit poor retrieval performance, with CORR values ranging from ~0.20 to ~0.30 (Fig. S5), which is primarily attributed to limitations in the spatial representativeness of the training data. The deep-learning module was trained exclusively on a long-term dataset from a single site in Beijing, which is insufficient to capture the spatial variability in emission intensity, as well as local meteorological and geographical conditions across the broader NCP. As a result, the spatial extrapolation capability of the deep-learning module is constrained. Although the retrieval framework can retrieve $PM_{2.5}$ chemical component concentrations at spatially distributed lidar sites, future work should incorporate long-term datasets from varying locations to enhance spatial generalization and extrapolation performance.*"

**A3**: In response to the reviewer's suggestions, we have added a map to show the training and independent test sites (**Fig. S1**). Notably, 39 independent testing sites with obvious outliers and missing values in the original manuscript were removed in the

revised manuscript (the number of remaining sites is 23). And we have added **Fig. S4** and **Fig. S5** to present statistical metrics by scatterplots and error distribution histograms across all sites and problematic sites (CORR <0.5).

[Figure]

**Figure S1**: *Spatial distribution of Beijing and spatially independent testing sites. Beijing site provides 18-month datasets for the training, validation and temporally independent testing of the deep-learning module. Other 23 sites in North China Plain (NCP) provide 8-day datasets for the spatially independent testing of the final retrieval. The geographic basemap is hosted by Esri (https://www.esri.com/en-us/home).*

[Figure]

***Figure S4****: Scatterplots of the retrievals (µg m⁻³) versus the observations (µg m⁻³) with probability density (%) for NH₄⁺, NO₃⁻, SO₄²⁻, OM and BC across 23 spatially independent testing sites (a1-a5). The dotted grey lines represent the 2:1, 1:1, and 1:2 lines, and the solid red line represents the fitted regression line. CORR represents the correlation coefficient, and RMSE represents root mean square error. Probability distributions of error (observations minus retrievals, µg m⁻³) for NH₄⁺, NO₃⁻, SO₄²⁻, OM and BC across 23 spatially independent testing sites (b1-b5). ME: Mean Error; SD: Standard Deviation.*

[Figure]

***Figure S5****: Scatterplots of the retrievals (µg m⁻³) versus the observations (µg m⁻³) with probability density (%) for NH₄⁺, NO₃⁻, SO₄²⁻, OM and BC across problematic (CORR <0.5) spatially independent testing sites (a1-a5). The dotted grey lines represent the 2:1, 1:1, and 1:2 lines, and the solid red line represents the fitted regression line. CORR represents the correlation coefficient, and RMSE represents root mean square error. Probability distributions of error (observations minus retrievals, µg m⁻³) for NH₄⁺, NO₃⁻, SO₄²⁻, OM and BC across 23 spatially independent testing sites (b1-b5). ME: Mean*

*Error; SD: Standard Deviation.*

**6) Lack of uncertainty quantification**

*Given the physics-constrained framing, the model should also provide uncertainty estimates, or at minimum a discussion of uncertainty propagation. Possible approaches include: Ensemble modeling, Monte Carlo dropout, error propagation from lidar extinction + physics constraints. Uncertainty bounds would greatly strengthen confidence in profile retrievals.*

**Authors' response:**

We thank the reviewer for this crucial suggestion, and we agree that providing uncertainty assessment is essential for a robust and trustworthy retrieval framework. In response to the reviewer's suggestion, we have added a dedicated section to discuss the uncertainties. The uncertainty sources primarily include the hyperparameters of the deep learning module, the input data, and the physical constraints.

**a. Uncertainty induced by the hyperparameters**

Hyperparameters largely determine the deep neural network architecture, training performance, training efficiency, and generalization capability, critically shaping the accuracy of the mapping between aerosol extinction coefficients and $PM_{2.5}$ chemical composition concentrations. To mitigate the hyperparameter-induced uncertainty, we employed the Bayesian optimization to identify the optimal set of hyperparameters that minimized the average 10-fold CV mean absolute error (MAE). Consequently, the hyperparameter-induced uncertainty will not be discussed further.

**b. Uncertainty induced by the input data**

The data of auxiliary features (e.g. meteorological parameters and aerosols) used for deep learning and physics-constrained optimization, are obtained from the global reanalysis products (e.g. ERA5 and CAMS). These grid data with relatively coarse spatial resolutions would yield errors when interpreting the local features at a specific observational site. Therefore, acquiring the vertical observational data of these auxiliary features for retrieval input can effectively mitigate the uncertainty induced by the input data.

**c. Uncertainty induced by the physical constraints**

The physical constraint function currently adopted is derived from the revised

IMPROVE equation (Pitchford et al., 2007). When compared with light scattering coefficients ($B_{sp}$) measured by nephelometers at seven IMPROVE sites between 2003 and 2012, this equation tends to overestimate $B_{sp}$ in the lower quintile and underestimate them in the upper quintile (Lowenthal and Kumar, 2016). Although the IMPROVE equation can estimate aerosol extinction coefficients using chemical composition concentrations, mass scattering efficiencies, and hygroscopic properties, its inherent estimation biases introduce additional uncertainty when it is applied as a physical constraint.

In summary, the uncertainty related to hyperparameters has been mitigated through Bayesian optimization, thus ***Section 3.4*** focuses on uncertainties arising from input data and physical constraints.

Section 3.4, Limitations and uncertainties: "*The deep learning module in our retrieval framework can establish a powerful mapping between optical and meteorological features and PM$_{2.5}$ chemical species, and physics-based explicit constraints can enhance the reliability and expandability of the mapping relationships. However, several limitations and sources of uncertainty remain and should be acknowledged when interpreting the results and extending the framework to broader applications.*

*First, the spatial scope of the training data is predominantly restricted to the NCP region. Expanding the retrieval framework with data from more diverse geographical locations is necessary to improve its global transferability. Second, the current retrieval framework primarily relies on extinction coefficients at a wavelength of 532 nm, exhibiting dependence on specific lidar instruments. Future retrieval framework should focus on integrating diverse optical features from additional wavelengths to enhancing adaptability and transferability. Third, the auxiliary input data used in both the deep learning module and the physics-constrained optimization are obtained from global reanalysis products, which may not fully capture local atmospheric conditions at specific observational sites, thereby introducing representativeness errors into the retrievals. Acquiring the vertical observational data for these auxiliary features can effectively mitigate the uncertainty induced by the input data. Fourth, the IMPROVE equation applied as an external physical constraint may introduce additional*

*uncertainty into the retrievals due to its systematic estimation biases (Lowenthal and Kumar, 2016). Moreover, since the IMPROVE equation was applied as an external physical constraint to optimize the retrievals of PM$_{2.5}$ chemical components, the machine learning model itself was not intrinsically constrained by physical principles during its training. Future work could incorporate an internal physical constraint into the machine learning model to improve its physical interpretability by formulating a hybrid loss function for training that combines the traditional data-fitting term with a physical term. Finally, long-term acquisition of independent vertical profiling data from both tower-based and aircraft-based campaigns is essential for a comprehensive assessment of the robustness of the vertical retrievals with respect to varying sites, aerosol types, and seasons.*"

**Reference**

Lowenthal, D. H. and Kumar, N.: Evaluation of the IMPROVE Equation for estimating aerosol light extinction. J. Air Waste Manage., 66, 726-737. https://doi.org/10.1080/10962247.2016.1178187, 2016.

Pitchford, M., Malm, W., Schichtel, B., Kumar, N., Lowenthal, D., and Hand, J.: Revised Algorithm for Estimating Light Extinction from IMPROVE Particle Speciation Data, J. Air Waste Manage., 57, 1326-1336, https://doi.org/10.3155/1047-3289.57.11.1326, 2007.

*3 Minor Comments and Suggestions:*

*1) If possible, include an ablation study comparing architectures (CNN, BiLSTM, CNN+BiLSTM, transformer) to justify the chosen hybrid design.*

**Authors' response:**

We thank the reviewer for the valuable suggestion. We fully agree that a systematic ablation study is essential to justify our chosen hybrid architecture. Our decision to employ the CNN-BiLSTM-Attention framework is informed by established findings in related fields. For instance, our previous work demonstrated the superiority of a hybrid CNN-BiLSTM over a standalone LSTM for interpreting PM$_{2.5}$ chemical components (Li et al., 2025). This conclusion is further supported by comparative studies in other domains, such as geoscience and finance, where the hybrid CNN-BiLSTM-Attention architecture was shown to outperform alternatives like CNN, LSTM, BiLSTM and

BiLSTM-Attention (Kavianpour et al., 2023; Ma et al., 2022; Shan et al., 2021; Zhang et al., 2023).

The primary goal of this study is to establish a novel framework that maps lidar vertical extinction coefficients to $PM_{2.5}$ chemical component concentrations using deep learning with an external physical constraint. In response to the reviewer's suggestion, the rationale behind the selection of our hybrid architecture has been further elaborated in the Introduction of the revised manuscript.

Section 2.2.2, Line 182-186: "…*The CNN and BiLSTM layers, coupled with the Attention Mechanism (AM), are designed to effectively capture the multivariate and temporal characteristics in the training data, thereby establishing a robust nonlinear mapping between the input and output features. The hybrid CNN-BiLSTM-AM architecture consistently outperforms single-architecture models in predictive tasks, as evidenced by numerous studies. (Kavianpour et al., 2023; Ma et al., 2022; Shan et al., 2021; Zhang et al., 2023)…*"

**Reference**

Kavianpour, P., Kavianpour, M., Jahani, E., and Ramezani, A.: A CNN-BiLSTM model with attention mechanism for earthquake prediction, J. Supercomput., 79, 19194-19226, https://doi.org/10.1007/s11227-023-05369-y, 2023.

Li, H., Yang, T., Du, Y., Tan, Y., Wang, Z.: Interpreting hourly mass concentrations of $PM_{2.5}$ chemical components with an optimal deep-learning model, J. Environ. Sci., 151, 125-139, https://doi.org/10.1016/j.jes.2024.03.037, 2025.

Ma, T., Xiang, G., Shi, Y., Liu, Y.: Horizontal in situ stresses prediction using a CNN-BiLSTM-attention hybrid neural network, Geomech. Geophys. Geo-energ. Geo-resour. 8, 152, https://doi.org/10.1007/s40948-022-00467-2, 2022.

Shan, L., Liu, Y., Tang, M., Yang, M., Bai, X.: CNN-BiLSTM hybrid neural networks with attention mechanism for well log prediction, J. Petrol. Sci. Eng., 205, 108838, https://doi.org/10.1016/j.petrol.2021.108838, 2021.

Zhang, J., Ye, L., Lai, Y.: Stock price prediction using CNN-BiLSTM-Attention model, Mathematics, 11, 1985, https://doi.org/10.3390/math11091985, 2023.

*2) Clarify the meaning of "estimated and observed extinction coefficients" (line 192). Does "estimated" refer to IMPROVE-derived extinction?*

**Authors' response:**

We thank the reviewer for pointing out this lack of clarity. In response to the

reviewer's suggestion, we have clarified the meaning of "estimated and observed extinction coefficients" in the revised manuscript.

Section 2.2.3, Line 267-269: "*…The NSGA-II algorithm is utilized to determine the optimal scale factors by solving the multi-objective function that simultaneously enhances the correlation and reduces the discrepancy between the IMPROVE-estimated and lidar-observed extinction coefficients.*"

*3) Define all acronyms at first use; ensure figure captions are self-contained.*

**Authors' response:**

We thank the reviewer for highlighting these important editorial points, which are essential for manuscript clarity and accessibility. Following the reviewer's suggestions, we have addressed both points in the revised manuscript.

Figure 1: "***Figure 1**: Remote-sensing retrieval framework for vertical distribution of five $PM_{2.5}$ chemical components ($NH_4^+$, $SO_4^{2-}$, $NO_3^-$, OM and BC). (U: U-component wind; V: V-component wind; T: Temperature; RH: Relative Humidity; q: Specific Humidity; w: Vertical Velocity; Z: Geopotential; $\boldsymbol{\sigma_{ext,532}}$ : Aerosol Extinction Coefficient at 532 nm; CNN: Convolutional Neural Network; ReLU: Rectified Linear Unit; FC: Fully Connected; BiLSTM: Bidirectional Long Short-Term Memory; IMPROVE: Interagency Monitoring of Projected Visual Environment; NSGA-II: Non-dominated Sorting Genetic Algorithm II).*"

Figure 2: "***Figure 2**: Brief workflow of NSGA-II (A: the parent population; B: the offspring population; C: the new population; P: the Pareto front).*"

Figure 3: "***Figure 3**: Scatterplots of the simulations ($\mu g\ m^{-3}$) versus the observations ($\mu g\ m^{-3}$) with probability density (%) for $NH_4^+$, $NO_3^-$, $SO_4^{2-}$, OM and BC during the 10-fold cross-validation process (a1-a5) and temporally independent testing process (b1-b5). The dotted grey lines represent the 2:1, 1:1, and 1:2 lines, and the solid red line represents the fitted regression line. CORR represents the correlation coefficient, and RMSE represents root mean square error.*"

Figure 4: "***Figure 4**: Weekly-smoothed variations in the retrieved and observed concentrations ($\mu g\ m^{-3}$) of $NH_4^+$ (a1), $NO_3^-$ (a2), $SO_4^{2-}$ (a3), OM (a4) and BC (a5) in*

*2021. (b) same as (a1-a5) but for $PM_{2.5}$ in 2017. (c) same as (a1-a5) but for $PM_{2.5}$ in 2018. (d) same as (a1-a5) but for $PM_{2.5}$ in 2024. CORR represents the correlation coefficient, RMSE represents root mean square error.*"

Figure S3: "***Figure S3**: Probability distributions of error (observations minus simulations, $\mu g\ m^{-3}$) for $NH_4^+$, $NO_3^-$, $SO_4^{2-}$, OM and BC during the 10-fold cross-validation phase (a1-a5) and during the temporally independent testing phase (b1-b5). ME: Mean Error; SD: Standard Deviation.*"

Figure S4: "***Figure S4**: Scatterplots of the retrievals ($\mu g\ m^{-3}$) versus the observations ($\mu g\ m^{-3}$) with probability density (%) for $NH_4^+$, $NO_3^-$, $SO_4^{2-}$, OM and BC across 23 spatially independent testing sites (a1-a5). The dotted grey lines represent the 2:1, 1:1, and 1:2 lines, and the solid red line represents the fitted regression line. CORR represents the correlation coefficient, and RMSE represents root mean square error. Probability distributions of error (observations minus retrievals, $\mu g\ m^{-3}$) for $NH_4^+$, $NO_3^-$, $SO_4^{2-}$, OM and BC across 23 spatially independent testing sites (b1-b5). ME: Mean Error; SD: Standard Deviation.*"

Figure S5: "***Figure S5**: Scatterplots of the retrievals ($\mu g\ m^{-3}$) versus the observations ($\mu g\ m^{-3}$) with probability density (%) for $NH_4^+$, $NO_3^-$, $SO_4^{2-}$, OM and BC across problematic (CORR <0.5) spatially independent testing sites (a1-a5). The dotted grey lines represent the 2:1, 1:1, and 1:2 lines, and the solid red line represents the fitted regression line. CORR represents the correlation coefficient, and RMSE represents root mean square error. Probability distributions of error (observations minus retrievals, $\mu g\ m^{-3}$) for $NH_4^+$, $NO_3^-$, $SO_4^{2-}$, OM and BC across 23 spatially independent testing sites (b1-b5). ME: Mean Error; SD: Standard Deviation.*"

*4) The manuscript describes z-score normalization but not how denormalization is performed. Why use aircraft-based measurements for denormalization instead of lidar-derived extinction? Explain and quantify the sensitivity.*

**Authors' response:**

    We sincerely thank the reviewer for this critical question, which has allowed us to correct a significant point of confusion in our original description and to clarify our

methodology. We wish to clarify that aircraft-based measurements should not be used for the denormalization step. This was a serious misstatement in our original manuscript, for which we apologize. The mean and standard deviation statistics used for inverse Z-score transformation must be derived from the original training set. In response to the reviewer's suggestion, we have revised the entire manuscript by using the correct denormalization parameters and added revised description of denormalization in the revised manuscript.

Section 2.2.1, Line 157-169: "

*Step 1. The multi-source input datasets undergo matching across spatiotemporal and vertical dimensions. All input and output data are uniformly time-resolved to hourly intervals, while vertical data are uniformly vertically resolved into 10 layers ranging from 50 m to 3 km.*

*Step 2. The input data of the deep learning module are normalized by Z-score normalization to stabilize the training process, accelerate training convergence, and enhance model robustness (Al-Faiz et al., 2018; Cabello-Solorzano et al., 2023).*

*Step 3. Training deep learning module by using the normalized surface-level input data.*

*Step 4. Generating the normalized concentrations of the five $PM_{2.5}$ chemical components at each vertical layer by feeding the normalized height-level input data into the deep learning module.*

*Step 5. Denormalizing the deep-learning output by using the inverse Z-score transformation, with the mean and standard deviation statistics derived from the original training set, thereby recovering the physical mass concentration unit ($\mu g\ m^{-3}$).*

*Step 6. Optimizing the denormalized deep learning output through implementing an external physics constraint to obtain the high-quality vertical concentration profiles of the five $PM_{2.5}$ chemical components. Repeat steps 4-6 until the retrieval task is complete.*"

Section 2.2.3, Line 244-245: "*The normalized vertical profiles of $PM_{2.5}$ chemical components generated by the deep learning module are denormalized by the statistical characteristics of the initial input data of the surface-level observations…*"

*5) The scaling procedure using the ratio of in situ to aircraft PM$_{2.5}$ ("initially scaled...")
is ambiguous. Provide a clear mathematical expression and discuss whether this
introduces bias.*

**Authors' response:**

We thank the reviewer for this important request for clarification. As presented in
the replies of **Minor Comments #4**, aircraft-based measurements should not be used
for the denormalization step. We have corrected the corresponding contents in the
revised manuscript, including the figures and results. The revised version can be found
in the uploaded file ***manuscript_with track changes.docx/pdf***.

Section 2.2.3, Line 244-249: "*The normalized vertical profiles of PM$_{2.5}$ chemical
components generated by the deep learning module are denormalized by the statistical
characteristics of the initial input data of the surface-level observations. To reduce the
retrieval error induced by the inherent extrapolation limitations of deep learning
modules, a physics-constrained optimization scheme is incorporated into the retrieval
framework based on a revised Interagency Monitoring of Projected Visual Environment
(IMPROVE) Equation (Pitchford et al., 2007) and Non-dominated Sorting Genetic
Algorithm II (NSGA-II) (Verma et al., 2021).*"

*6) The description of the attention layer lacks physical interpretation. Is attention
purely data-driven, or does physics guide attention weights? If physics influences
attention, show how.*

**Authors' response:**

We thank the reviewer for prompting us to provide a clearer interpretation of the
attention mechanism within our retrieval framework.

In our framework, we employ a purely data-driven Channel Attention Mechanism
positioned between the CNN and BiLSTM layers. First, the multivariate features
learned by the convolutional layers are compressed through global average pooling.
Second, the compressed features are transmitted into fully connected layers with a
sigmoid activation function to generate a set of channel attention weights. Third, the
learned weights are utilized to re-scale the original feature channels from the
convolutional layers through element-wise multiplication.

In response to the reviewer's suggestion, we revised the methodology description to clearly state the data-driven nature of the attention layer and its position within the broader physics-constrained framework.

Section 2.2.2, Line 200-207: "*The attention mechanism layer is incorporated with CNN to amplify the weight of key information and mitigate the interference of redundant information, leading to an enhancement in the quality of the CNN output (Wang and Zhang, 2025). The attention mechanism is inspired by the ability of human vision to selectively focus on key information (Guo et al., 2022). Our retrieval framework integrates a data-driven channel attention mechanism, which rescales the original feature channels from the convolutional layers through element-wise multiplication using learned attention weights, thereby enhancing the importance of key features and reduce the interference of irrelevant features. The attention weights are generated by the FC layers with a sigmoid activation function (Eq. (2)) and then performs Schur product operation with CNN multivariate output (Eq. (3)).*"

*7) Figure 1 needs explicit legends for color boxes/arrows and clear annotation of all inputs and outputs.*

**Authors' response:**

In response to the reviewer's suggestions, we have revised ***Fig .1*** as follows.

[Figure]

***Figure 1***: *Remote-sensing retrieval framework for vertical distribution of five PM$_{2.5}$ chemical components (NH$_4^+$, SO$_4^{2-}$, NO$_3^-$, OM and BC). (U: U-component wind; V: V-component wind; T: Temperature; RH: Relative Humidity; q: Specific Humidity; w: Vertical Velocity; Z: Geopotential; $\boldsymbol{\sigma_{ext,532}}$: Aerosol Extinction Coefficient at 532 nm; CNN: Convolutional Neural Network; ReLU: Rectified Linear Unit; FC: Fully Connected; BiLSTM: Bidirectional Long Short-Term Memory; IMPROVE: Interagency Monitoring of Projected Visual Environment; NSGA-II: Non-dominated Sorting Genetic Algorithm II).*

*8) Figure 5a does not effectively show differences between datasets. Consider: a) scatterplots colored by site with standard deviations, b) an additional plot showing error distribution histograms for each of the five components.*

**Authors' response:**

    We thank the reviewer for the suggestions, and we have added ***Fig. S4*** and ***Fig.***

*S5* to provide supplementary information for Fig. 5a by using scatterplots and error distribution histograms. ***Fig. S4*** and ***Fig. S5*** are presented in the replies of **Major Comments #5 A3**.

*9) Include full training hyperparameters: batch size, learning rate, optimizer, epochs, early stopping criteria, normalization statistics.*

**Authors' response:**

In response to the reviewer's suggestions, we have added a **Table S2** into the supplement to present full training hyperparameters.

**Table S2**. Optimal hyperparameters of the deep learning module.

| Hyperparameter | Decision space | Optimal values |
|---|---|---|
| Initial learning rate | $[10^{-5}\ 10^{-3}]$ | $4.71 \times 10^{-4}$ |
| Factor for $L_2$ regularization | $[10^{-10}\ 10^{-2}]$ | $1.54 \times 10^{-4}$ |
| Decay rate of gradient moving average | $[0.8\ 0.98]$ | 0.80 |
| Decay rate of squared gradient moving average | $[0.8\ 0.99]$ | 0.81 |
| Number of filters | 1 $[8\ 64]$ | 44 |
| | 2 $[8\ 64]$ | 34 |
| Size of filters | 1 $[3\ 16]$ | 6 |
| | 2 $[3\ 16]$ | 10 |
| Number of layers | $[1\ 4]$ | 2 |
| Number of hidden units | $[60\ 200]$ | 61 |
| Maximum of Epochs | \ | 100 |
| Size of mini-batch | \ | 64 |
| Dropout value | \ | 0.25 |
| Solver | \ | adam |
| Num of cross-validation folds | \ | 10 |

Section 2.2.2, Line 238-239: "*...The number of optimization iteration is set to 30 and the final optimal settings of model hyperparameters are presented in Table S2 of the supplement.*"